# 🧁 MUFFIN: CURATING MULTI-FACETED INSTRUCTIONS FOR IMPROVING INSTRUCTION-FOLLOWING

**Renze Lou**[†]  **Kai Zhang**[◇]  **Jian Xie**[‡]  **Yuxuan Sun**[♯]
**Janice Ahn**[†]  **Hanzi Xu**[♣]  **Yu Su**[◇]  **Wenpeng Yin**[†]

[†]The Pennsylvania State University; [◇]The Ohio State University;
[‡]Fudan University; [♯]Westlake University; [♣]Temple University
{renze.lou, wenpeng}@psu.edu

## ABSTRACT

In the realm of large language models (LLMs), enhancing instruction-following capability often involves curating expansive training data. This is achieved through two primary schemes: i) `Scaling-Inputs`: Amplifying (input, output) pairs per task instruction, aiming for better instruction adherence. ii) `Scaling Input-Free Tasks`: Enlarging tasks, each composed of an (instruction, output) pair without requiring a separate input anymore. However, LLMs under `Scaling-Inputs` tend to be overly sensitive to inputs, leading to misinterpretation or non-compliance with instructions. Additionally, `Scaling Input-Free Tasks` demands a substantial number of tasks but is less effective in instruction-following when dealing with instances in `Scaling-Inputs`. This work introduces MUFFIN, a new scheme of instruction-following dataset curation. Specifically, we automatically `Scale Tasks per Input` by diversifying these tasks with various input facets. Experimental results across four zero-shot benchmarks, spanning both `Scaling-Inputs` and `Scaling Input-Free Tasks` schemes, reveal that LLMs, at various scales, trained on MUFFIN generally demonstrate superior instruction-following capabilities compared to those trained on the two aforementioned schemes.[1]

## 1 INTRODUCTION

With advancements in pre-training techniques, large language models (LLMs) can, to some extent, tackle diverse unseen tasks guided by textual instructions (Radford et al., 2019; Brown et al., 2020). This capability, known as *Instruction-Following*, is pivotal for developing unified versatile LLMs. Instruction-tuning, training LLMs to generate desired responses following given instructions for enhanced instruction-following capacity, has garnered increased attention in the community (Min et al., 2022; Chung et al., 2022; Longpre et al., 2023; Lou et al., 2023).

The construction of datasets is crucial in instruction-tuning (Wang et al., 2023a; Zhou et al., 2023). Existing approaches primarily adopt two strategies for constructing these datasets: (i) `Scaling-Inputs` — gathering a vast set of training tasks, each accompanied by an instruction, and then amplifying the (input, output) pairs for each task (Mishra et al., 2022b; Sanh et al., 2022; Wei et al., 2022; Wang et al., 2022). The model is trained to produce distinct outputs for various inputs under the same instruction. However, this approach tends to render the model excessively sensitive to inputs, often resulting in misinterpretation or non-compliance with explicit instruction requirements (Webson & Pavlick, 2022; Mishra et al., 2022a) like "··· *generate less than five words*", and suboptimal learning efficiency (Ivison et al., 2022; Deb et al., 2022). (ii) `Scaling Input-Free Tasks` — collecting task instructions that can be answered without additional inputs, e.g., "*give the name of the highest mountain in the world*", and expanding the (instruction, output) training pairs (Wang et al., 2023b; Xu et al., 2023a). Despite the intuitive alignment with human-assistance objectives, covering a wide range of diverse tasks and aiding in daily queries, the input-free nature

---

[1]All the code and data are available at our project page: https://renzelou.github.io/Muffin/

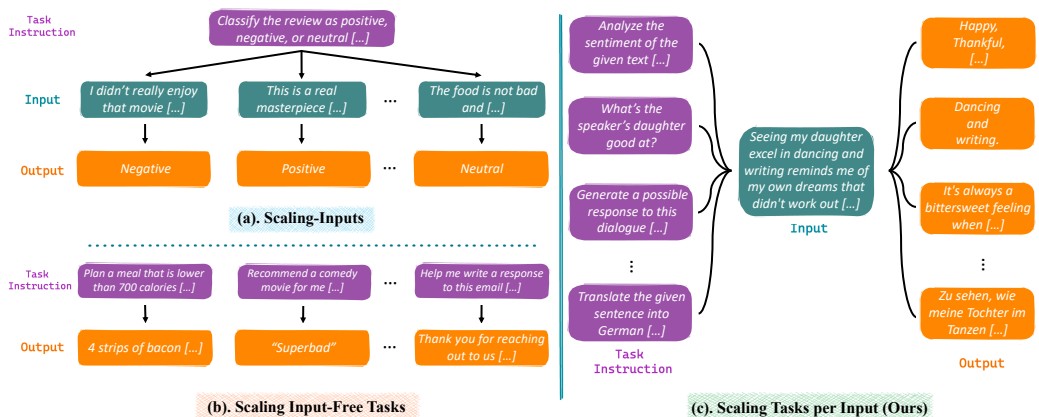

Figure 1: Three different paradigms for designing instruction-following datasets.

of the `Scaling Input-Free Tasks` paradigm makes the resulting LLMs less effective in handling traditional NLP tasks where instructions are accompanied by supplementary inputs.

In this study, we introduce a novel approach to curate instruction-following datasets, termed `Scaling Tasks per Input`, as illustrated in Figure 1(c). Instead of amplifying the task's input-output set in `Scaling-Inputs` or enlarging input-free tasks in `Scaling Input-Free Tasks`, the `Scaling Tasks per Input` paradigm introduces task diversification for each input. Consequently, models are trained to adapt outputs based on specific instructions related to the input, thus enhancing the instruction-following capacity of LLMs.

Two challenges in implementing `Scaling Tasks per Input`: ($\mathcal{C}_1$) Designing diverse tasks for the same input and ($\mathcal{C}_2$) Balancing classification and generation categories in the resulting dataset. Addressing $\mathcal{C}_1$, we propose two strategies to automatically synthesize varied tasks for each input. 1) *Instruction Brainstorm*: Enhancing task diversity and instruction-input relevance using an input-facet-oriented instruction generation approach. Instead of relying solely on existing human instructions as "demonstrations" for task brainstorming (Wang et al., 2023b; Honovich et al., 2022), we employ an input-facet-oriented procedure. LLMs identify diverse textual facets of the input, considering each facet as a "hint" to generate related instructions. 2) *Instruction Rematching*: Reusing high-quality task instructions from human-crafted datasets and determining their relevance to a given input. Diverse instructions for each input are collected, and LLMs annotate the output for each (instruction, input) pair, followed by filtering of raw results. Regarding $\mathcal{C}_2$, recognizing LLMs' inclination to produce more generation tasks than classification tasks, we propose a straightforward yet effective method to expand classification tasks. Our resulting dataset is named 🧁MUFFIN (**Mu**lti-**F**aceted **In**struction), the first instruction-following dataset aligning with `Scaling Tasks per Input`.

In experiments, we train LLMs of 3B&11B and evaluate them on four widely-used zero-shot benchmarks, i.e., SuperNI-Test (Wang et al., 2022), MMLU (Hendrycks et al., 2021a), T0-Eval (Sanh et al., 2022), and BBH (Suzgun et al., 2022). Automatic evaluations reveal that LLMs trained on our MUFFIN exhibit superior performance on three of the four benchmarks compared to models trained with various prior datasets from both `Scaling-Inputs` and `Scaling Input-Free Tasks` paradigms. Comprehensive human evaluation and in-depth analyses further affirm the effectiveness of our `Scaling Tasks per Input` paradigm and MUFFIN in improving the instruction-following capacities of LLMs.

To sum up, our main contributions are three-fold:

- We propose a brand-new paradigm for crafting instruction-following datasets — `Scaling Tasks per Input`. It enforces the model to follow various instructions in an input-controlling way.

- We develop a novel instruction synthesis framework that takes into account inputs' various facets. It increases task diversity (per input) and instruction-input relevance simultaneously. We publicly release our dataset MUFFIN and data construction code to benefit future research.

- Our MUFFIN enhances instruction-following capabilities of LLMs across scales, outperforming prior datasets constructed using various paradigms on standard evaluation benchmarks.

## 2 RELATED WORK

Initially, LLMs excel at following prompts — often concise cloze questions (Radford et al., 2019; Schick & Schütze, 2021a;b). By converting original task inputs into prompts, LLMs can achieve zero-shot performance across various tasks without parameter updates (Liu et al., 2023). However, the effectiveness of prompts heavily relies on LLMs, potentially leading to over-reliance on large models and vulnerability in robustness (Bach et al., 2022; Khashabi et al., 2022; Gu et al., 2022). To create improved instruction-following LMs, prior research turned to *Instruction Tuning* — training LLMs on extensive upstream tasks with instructions and then generalizing to downstream unseen tasks using new instructions (Sanh et al., 2022; Wei et al., 2022; Ouyang et al., 2022; Chung et al., 2022; Yin et al., 2022; Longpre et al., 2023). Consequently, collecting diverse and high-quality upstream training datasets becomes a pivotal step in successful instruction tuning (Wang et al., 2023a; Lou et al., 2023). We then discuss two main data curation approaches.

**Human Annotated Data.** The traditional instruction data creation relies on extensive human annotations (Xu et al., 2022b; Srivastava et al., 2022; Conover et al., 2023). For instance, PUBLIC POOL OF PROMPTS (P3) (Sanh et al., 2022) and FLAN (Wei et al., 2022) curated multi-task datasets with various task categories, leveraging human expertise to design prompt templates. Wang et al. (2022) introduced SUPER NATURAL INSTRUCTIONS (SUPERNI) by collecting 1.6k NLP tasks from the NATURAL INSTRUCTIONS dataset (Mishra et al., 2022b), employing 88 experts to brainstorm novel tasks. Despite the quality, human annotation is effort-intensive and time-consuming, especially for devising diverse and complex textual tasks.

**LLM Synthetic Data.** Recent research favors leveraging the creative capabilities of LLMs, like ChatGPT (OpenAI, 2022) or GPT-4 (OpenAI, 2023), over human input for creating instruction-following datasets (Xu et al., 2023b; Geng et al., 2023; Chiang et al., 2023; Kim et al., 2023; Ding et al., 2023). Approaches such as SELF-INSTRUCT (Wang et al., 2023b) and UNNATURAL INSTRUCT (Honovich et al., 2022) utilized human-annotated instructions as demonstrations to guide LLMs in devising novel tasks and increasing task diversity. ALPACA (Taori et al., 2023) and ALPACA-GPT4 (Peng et al., 2023) utilized more powerful LLMs to enhance data quality. Another line of research uses free-form texts as "seeds" to assist LLMs in brainstorming task instructions (Wu et al., 2023). For instance, LONGFORM (Köksal et al., 2023) created instructions based on lengthy output text to improve long text generation abilities of LLMs; WIZARDLM (Xu et al., 2023a) employed an instruction evolution paradigm to increase seed instruction complexity; DYNOSAUR (Yin et al., 2023) repurposed existing input-output pairs in NLP datasets to stimulate new instructions and reduce annotation costs. In contrast, our instruction-brainstorm pipeline uses the task input as the "seed", with all new instructions aimed at processing this input into different outputs.

## 3  🧁 MUFFIN CURATION

This section elaborates on the key modules for constructing MUFFIN: input collection from diverse sources (§ 3.1), addressing $\mathcal{C}_1$ by generating input-oriented diverse tasks (§ 3.2), followed by output annotation and filtering steps, and tackling $\mathcal{C}_2$ by controlling the classification-generation balance (§ 3.3). The complete pipeline is illustrated in Figure 2. We show the API usage and costs in Table 8.

### 3.1  INPUT COLLECTION

We sample inputs from two distinct sources to ensure diversity (sampling details in Appendix A):

**SUPERNI** (Wang et al., 2022) is a human-annotated dataset encompassing 1,600+ NLP tasks across diverse categories, sourced from existing benchmarks or created by human experts, implying a remarkable input diversity of SUPERNI. Therefore, we randomly select inputs from the training tasks of SUPERNI as our input text source (only inputs, no outputs are sampled at this stage).

**Dolma** (Soldaini et al., 2023) is a vast corpus used for pretraining, covering free-form texts from four domains: *web content*, *academic publications*, *code*, and *encyclopedic materials*.[2] Texts are randomly sampled based on domain sizes, prioritizing domains with fewer texts to ensure diversity.

---

[2]We exclude *book* domain texts due to their length and unsuitability as task inputs.

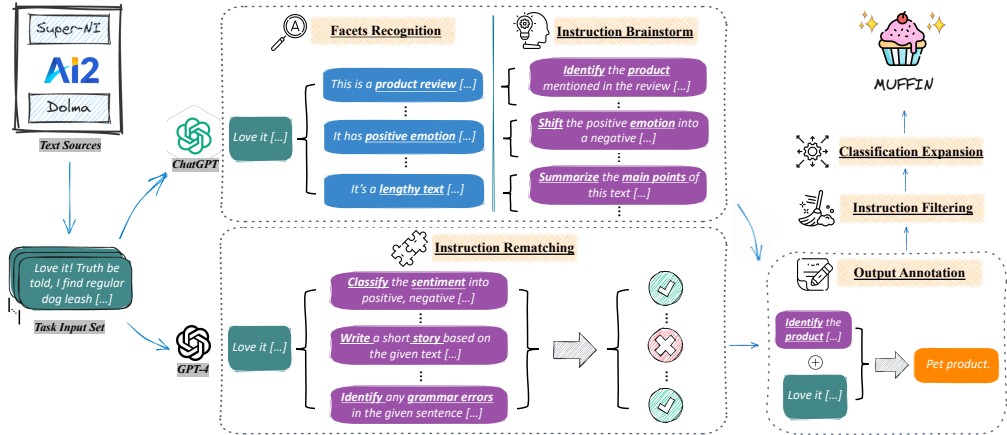

Figure 2: Data construction pipeline of 🧁 MUFFIN.

## 3.2 INSTRUCTION COLLECTION

After collecting a diverse set of task inputs, we design two strategies to generate task instructions: 1) **Instruction Brainstorm** — LLMs generate instructions based on the input, and 2) **Instruction Rematching** — reusing existing human-crafted instructions on the input, evaluated by LLMs.

**Instruction Brainstorm Based on Inputs' Facets.** A common approach for generating task instructions is to use existing human-written instructions as few-shot demonstrations to guide LLMs, as seen in prior works (Wang et al., 2023b; Honovich et al., 2022; Taori et al., 2023). However, our initial experiments revealed challenges in directly generating suitable instructions for the input using this conventional method — LLMs tended to produce unrelated tasks, emphasizing the importance of using specific facets (attributes) of the input to stimulate corresponding tasks.

To this end, in Figure 2, we introduce a two-step facet-based instruction brainstorming method. Firstly, we prompt ChatGPT to recognize the *textual facets* of the given input, aiming to identify as many facets as possible using an enumeration prompt (as shown in Table 3). Secondly, with each recognized facet as a *hint*, we instruct ChatGPT to brainstorm task instructions for the input. Similar to prior works (Wang et al., 2022; Honovich et al., 2022), we randomly sample three instructions from SUPERNI's training set as demonstrations to increase the validity of generated tasks (prompt details in Table 3 and Table 4).

**Instruction Rematching.** Our data collection pipeline aims to associate an input with diverse task instructions. Besides direct instruction synthesis by LLMs, another approach is to gather suitable instructions from existing human-annotated sets. Illustrated in Figure 2, we extract human-written instructions from SUPERNI's training set and employ LLMs (specifically GPT-4) for binary classification. The classification involves determining if the instruction can align with the input to form a valid task ("Yes" or "No"). Subsequently, we collect all matched instructions predicted by the LLMs for a given input. See Appendix B for more technical details of instruction rematching.

After obtaining (instruction, input) pairs, we use ChatGPT to annotate outputs, or mark them as "None" for instances that ChatGPT deems unanswerable (due to invalid instructions or mismatched input-instruction pairs). Table 6 shows the prompt. Next, we employ the following post-processing steps to filter out noisy instances: 1) *Overlapped Tasks* — for each input, remove its task instructions whose *ROUGE-L* similarity with any existing instructions is higher an empirical threshold (set at 0.7 in this work); 2) *Non-answerable Instances* — eliminate instances with outputs marked as "None".

## 3.3 CLASSIFICATION EXPANSION

Classification tasks, characterized by small fixed output spaces, are prevalent in real-world applications (Wang et al., 2019b;a; Sanh et al., 2022; Xu et al., 2022a). However, LLMs tend to generate significantly more generation instructions than classification ones (Yin et al., 2023), resulting in lower-than-expected generalization capabilities on classification tasks. Given the essence of classifi-

cation tasks being the selection of the most likely correct answer, we propose a straightforward and effective approach to expand the classification-oriented task instructions.

For a given (instruction, input, output) instance: 1) we first let ChatGPT generate additional "*wrong outputs*" that should be suboptimal compared to the correct output (see Table 7). 2) We combine the correct output with these wrong outputs to form the entire output space for the task, presented as options `A, B, C, etc.`, in a randomly shuffled order. Additionally, a sentence in the instruction specifies that the answer should be chosen from these options (`A, B, C`). This transformation helps convert original generation tasks into a classification formulation. To prevent answer letter bias, we use random alphabet, numbers, or special symbols (e.g., "`@, $, #`") as option letters.

We exclusively apply classification expansion to brainstormed instructions, as they are deficient in classification tasks. Simultaneously, we exclude the generation tasks with outputs' lengths exceeding a threshold (set at 100 words) due to difficulty in converting them into a classification paradigm. Ultimately, when presented with an original instance and its expanded classification version, we randomly select one (with equal probability) to be included in our MUFFIN, ensuring task balance.

## 4 DATA ANALYSES

**Statistics.** Table 1 lists the detailed statistics of the 68K (instruction, input, output) instances in MUFFIN. It is worth mentioning that some inputs will share the same instructions because of our "instruction rematching" mechanism. The length distribution of instructions, inputs, and outputs can be found in Figure 4 in Appendix.

**Quality.** To assess data quality, we randomly selected 200 instances from our dataset (200 random inputs, 1 random instruction per input). Two NLP graduate students were involved in the evaluation. Following Wang et al. (2023b), each annotator answered three questions for each instance: 1) determining if the instruction describes a valid task (A1) when only the instruction is provided, 2) assessing if the instruction appropriately matches

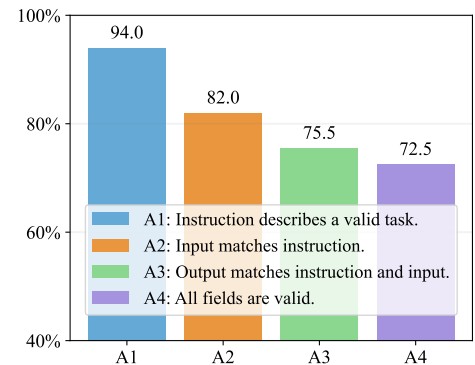

Figure 3: Human evaluation on the data quality. Both valid and invalid instances can be found in Table 13. A4 indicates the joint set of successful cases in A1, A2, and A3.

the input when only the instruction-input pair is presented (A2), and 3) evaluating if the output correctly responds to the instruction and input (A3). We also recorded instances where all three fields were correct (A4). Figure 3 shows the correct ratio for each question.[3] Our dataset achieved a 72.5% correct ratio on all three fields, significantly outdoing the 54% reported by Wang et al. (2023b).

**Diversity.** To analyze the task diversity of MUFFIN, we follow previous works (Wang et al., 2023b; Peng et al., 2023; Yin et al., 2023) using Berkeley Neural Parser (Kitaev & Klein, 2018) to phrase the task instructions in our dataset. We plot the top 20 most common root verbs of instructions along with their top 4 direct noun objects. As shown in Figure 5 in the Appendix, our MUFFIN shows a good instruction diversity, where most instructions focus on creative generation tasks.

Table 1: Statistics of MUFFIN.

| statistic | |
| --- | --- |
| # of inputs | 1,463 |
|   - # of inputs (from SuperNI) | 953 |
|   - # of inputs (from Dolma) | 510 |
| # of instructions | 56,953 |
|   - # of instructions by "rematching" (from SuperNI) | 574 |
|   - # of instructions (from brainstorm) | 33,720 |
|   - # of instructions (from classification expansion) | 22,659 |
| # of instructions per input | 46.48 |
| # of inputs per instruction[4] | 20.27 |
| # of (instruction, input, output) instances | 68,014 |
| ave. input length (in words) | 119.26 |
| ave. instruction length (in words) | 84.74 |
| ave. output length (in words) | 71.32 |

## 5 EXPERIMENTAL SETUP

We conduct a comprehensive evaluation that covers benchmarks and baselines with various paradigms. Here, we use `Scaling-Inputs` and `Scaling Input-Free Tasks` to denote the different prior

---

[3]The two annotators had similar correct ratios, with an average agreement of 83.3% for the three questions.

[4]We only report the "# of inputs per instruction" of rematching instruction data. As for brainstorming instruction, this number is almost equal to 1, because almost all the instructions generated by LLMs are different.

paradigms introduced in § 1. In addition, we use `Hybrid` to represent datasets constructed by converting `Scaling-Inputs` into `Scaling Input-Free Tasks` (i.e., concatenating original instruction and input to form an extended instruction without explicit input). Such datasets essentially belong to `Scaling-Inputs`, as they keep the same instruction while varying the inputs, but it is converted into an input-free style to cater to the models tuned on `Scaling Input-Free Tasks`.

**Evaluation Benchmarks.** In the zero-shot setting, we report on the following four benchmarks that are widely used in prior work. Please refer to Appendix I for prompt template details.

• **SuperNI-Test (Wang et al., 2022)** `Scaling-Inputs` The official test set of SUPERNI with 119 distinct NLP tasks spanning both classification and generation. Following previous works (Wang et al., 2022; Lou & Yin, 2023), we use the first 100 instances of each test task. To ensure the fully zero-shot generalization setting, we only utilize the task definition (instruction) of SUPERNI without any demonstrations. We use greedy decoding for all models tested on this benchmark. Official evaluation metrics: "*Exact-Match*" for classification, "*ROUGE-L*" for generation and overall performance.

• **MMLU (Hendrycks et al., 2021b)** `Scaling Input-Free Tasks` The Massive Multitask Language Understanding dataset (MMLU) covers extensive questions from 57 subjects with various difficulty levels. In total, there are 14,042 test instances. Each instruction in MMLU consists of a subject-related question along with four answer choices (i.e., "$\cdots$ answer with (A), (B), (C) or (D)"). We adapt two evaluation settings: 1) adopting greedy decoding generation and using *Exact-Match* score as metric, which is the most conventional choice (Wang et al., 2023a; Xu et al., 2023a); 2) utilizing rank classification[5] for decoding with *Accuracy* score, which is also a popular setting for the classification tasks and focuses more on evaluating the models' knowledge without suffering from the mismatching problem of LLMs' generation. As there are only (instruction, output) pairs in MMLU, we categorize it as `Scaling Input-Free Tasks` paradigm.

• **T0-Eval (Sanh et al., 2022)** `Hybrid` We utilize the held-out evaluation set of T0 (we refer to it as T0-Eval) as one of the zero-shot benchmarks. The original T0-Eval contains 23 different task categories, while some of these tasks are leaked in SUPERNI's training set. Therefore, we follow Honovich et al. (2022) keeping 6 task types for fair comparison, namely ANLI R1-R3, CB, COPA, and RTE, leading to 82 corresponding evaluation datasets. Due to the extreme instance number imbalance, we follow a similar procedure as Wang et al. (2022) using the first 100 instances per dataset to ensure the reproducibility. Similar to the MMLU, the tasks in T0-Eval are related to multi-choice classification. Hence, we follow previous works (Sanh et al., 2022; Honovich et al., 2022) utilizing *Exact-Match* and *Rank Classification Accuracy* as metrics.

• **BBH (Suzgun et al., 2022)** `Hybrid` The "hard" subset of BIG-Bench (Srivastava et al., 2022). BBH consists of 23 challenging tasks with a total of 6,511 instances. Similarly, we remove all the demonstrations and Chain-of-Thought prompts in BBH to ensure the zero-shot setting. We use greedy decoding and report the *Exact-Match* scores for all the models in our experiments, which is also the official setting adopted by a variety of previous works (Wang et al., 2023a; Mukherjee et al., 2023).

**Baselines.** We categorized previous works as `Scaling-Inputs` (UNNATURAL INSTRUCT and DYNOSAUR) or `Scaling Input-Free Tasks` (DOLLY, LONGFORM, ALPACA, ALPACA-GPT4, WIZARDLM, and SELF-INSTRUCT). All of these datasets are crafted using ChatGPT or GPT-4 (except DOLLY), which can be directly compared with our MUFFIN (*direct comparison*). As for the manually-created dataset, we report the performance of SUPERNI ( `Scaling-Inputs` ), which can be regarded as an upper bound of all these synthetic datasets (*indirect comparison*). Note that, the original training set of SUPERNI contains 757 tasks (excluding those non-English tasks and tasks that are leaked from the test set), while we follow Yin et al. (2023) using 681 tasks (90%) for training and considering the remaining 76 tasks (10%) as the validation set. In addition to the aforementioned baselines, we also report the performances of existing systems with larger numbers of parameter or tuning-data sizes, such as Flan-T5 (Wei et al., 2022) and T0 (Sanh et al., 2022). See Appendix H for more particular introductions and implementation details toward these baselines.

**Implementation Details.** We fine-tune models with various architectures on MUFFIN, including encoder-decoder T5-LM (Raffel et al., 2020) and decoder-only Llama (Touvron et al., 2023a). We provide more implementation details (e.g., hyper-parameters) in Appendix G.

---

[5]Choosing the answer option with the highest log-likelihood as the final prediction of the model.

Table 2: Results by T5-3B (*second block*) and T5-11B (*third block*). Each block contains indirect comparison (i.e., trained on in-distribution SUPERNI) and direct comparison (i.e., LLM-generated datasets). Scores that are indicated with [†] and [*] are adopted from Wang et al. (2023a) and Mukherjee et al. (2023). The *first block* in this table reports some larger models or models with more tuning data, as a reference. Since Flan-T5 is trained on some evaluation benchmarks, we don't report these scores (marked as "—"). The best scores under direct and indirect comparison settings are in **bold** and underlined, respectively. Those previous baselines/evaluation benchmarks are marked with different colors to represent their paradigms (see § 5). All scores here averaged over three runs.

| | Models | Data Size | SuperNI-Test EM (CLS) | SuperNI-Test ROUGE-L (GEN) | SuperNI-Test ROUGE-L (overall) | MMLU Rank ACC | MMLU EM | T0-Eval Rank ACC | T0-Eval EM | BBH EM | Average |
|---|---|---|---|---|---|---|---|---|---|---|---|
| | \multicolumn Larger Models / Vanilla Models / More Training Data (for reference) | | | | | | | | | | |
| Existing Systems | GPT-4 | / | 64.51 | 59.36 | 62.96 | / | 82.40[†] | / | 70.95 | 67.40[*] | / |
| | ChatGPT | / | 46.90 | 56.82 | 52.41 | / | 67.90[†] | / | 50.73 | 48.90[*] | / |
| | Flan-T5 (11B) | 14M | — | — | — | 49.97 | 45.97 | — | — | 40.92 | / |
| | Flan-T5 (3B) | 14M | — | — | — | 45.52 | 45.07 | — | — | 39.70 | / |
| | T0++ (11B) | 12M | 29.40 | 48.46 | 40.01 | 45.46 | 43.20 | 59.70 | 62.22 | 20.15 | 43.58 |
| | T0 (11B) | 50K | 24.08 | 41.15 | 32.85 | 42.87 | 40.08 | 57.52 | 60.00 | 29.04 | 40.95 |
| | T0 (3B) | 50K | 18.95 | 36.66 | 26.84 | 31.51 | 25.32 | 46.17 | 46.63 | 22.76 | 31.86 |
| | Vanilla T5 (11B) | 0 | 0.00 | 20.89 | 8.40 | 22.95 | 0.00 | 37.56 | 0.00 | 0.00 | 11.23 |
| | Vanilla T5 (3B) | 0 | 0.00 | 21.84 | 10.28 | 24.12 | 0.00 | 37.45 | 0.38 | 0.00 | 11.76 |
| | \multicolumn Human Annotated Data (indirect comparison) | | | | | | | | | | |
| T5-3B | SuperNI-Train | 68k | 35.46 | 48.01 | 43.25 | 38.42 | 36.97 | 49.65 | 48.73 | 19.60 | 40.01 |
| | \multicolumn Generated Data (direct comprison) | | | | | | | | | | |
| | Dolly | 15k | 0.49 | 34.32 | 14.52 | 23.05 | 0.00 | 39.84 | 6.78 | 5.71 | 15.59 |
| | LongForm | 23k | 0.00 | 33.58 | 11.29 | 23.07 | 0.00 | 39.68 | 0.62 | 3.84 | 14.01 |
| | Alpaca | 52k | 20.43 | 46.08 | 35.25 | 28.55 | 8.02 | 43.26 | 20.52 | 11.53 | 26.71 |
| | Alpaca-GPT4 | 52k | 11.72 | 41.84 | 27.49 | 23.89 | 0.00 | 41.51 | 14.14 | 8.50 | 21.14 |
| | WizardLM | 68k | 5.34 | 41.09 | 20.81 | 25.55 | 0.00 | 40.55 | 5.87 | 5.16 | 18.05 |
| | Self-Inst. | 82k | 29.59 | 43.70 | 36.87 | 27.11 | 23.55 | 41.74 | 38.57 | **20.53** | 32.71 |
| | Unnatural Inst. | 68k | 32.56 | 45.08 | 41.42 | 32.65 | 18.03 | 43.42 | 34.49 | 8.53 | 32.02 |
| | Dynosaur | 66k | 26.97 | 44.27 | 35.65 | 26.11 | 20.38 | 38.98 | 38.81 | 13.68 | 30.61 |
| | Muffin (Ours) | 68k | **33.84** | **49.52** | **42.63** | **36.27** | **29.75** | **46.35** | **44.45** | 14.25 | **37.13** |
| | \multicolumn Human Annotated Data (indirect comparison) | | | | | | | | | | |
| T5-11B | SuperNI-Train | 68k | 41.13 | 50.05 | 47.76 | 54.45 | 54.37 | 56.89 | 54.23 | 29.80 | 48.59 |
| | \multicolumn Generated Data (direct comprison) | | | | | | | | | | |
| | Dolly | 15k | 2.71 | 37.12 | 17.81 | 22.99 | 0.06 | 49.17 | 23.96 | 10.18 | 20.50 |
| | LongForm | 23k | 1.88 | 38.05 | 16.27 | 23.23 | 0.00 | 39.85 | 2.79 | 5.53 | 15.95 |
| | Alpaca | 52k | 25.36 | 47.74 | 39.62 | 30.17 | 8.10 | 54.48 | 34.90 | 9.28 | 30.21 |
| | Alpaca-GPT4 | 52k | 13.65 | 43.19 | 31.46 | 25.58 | 0.00 | 49.94 | 34.79 | 7.94 | 25.82 |
| | WizardLM | 68k | 4.81 | 40.43 | 21.26 | 24.63 | 0.01 | 45.10 | 6.44 | 4.79 | 18.43 |
| | Self-Inst. | 82k | 28.88 | 44.88 | 36.53 | 28.22 | 32.45 | 48.61 | 41.46 | **31.39** | 36.55 |
| | Unnatural Inst. | 68k | 41.11 | 47.46 | 45.54 | 34.38 | 22.39 | 43.40 | 41.91 | 12.84 | 36.13 |
| | Dynosaur | 66k | **42.02** | 47.53 | 46.42 | 27.60 | 24.96 | 42.85 | 43.39 | 9.22 | 35.50 |
| | Muffin (Ours) | 68k | 40.20 | **50.69** | **48.32** | **41.95** | **41.83** | **55.38** | **57.74** | 20.53 | **44.58** |

# 6 EXPERIMENTAL RESULTS

We report automatic evaluation in § 6.1, human evaluation in § 6.2 and in-depth analyses in § 6.3.

## 6.1 AUTOMATIC EVALUATION

Table 2 shows the main results of fine-tuning T5-3B and T5-11B on different datasets. Additionally, we add the results based on Llama2 in Table 15 to further enhance our conclusion.

Compared with the previous LLM-generated datasets, such as SELF-INSTRUCT, UNNATURAL IN-STRUCT, and DYNOSAUR, the models tuned on our MUFFIN consistently achieve better performance across 3 out of 4 benchmarks, under various metrics. Besides the high quality and diversity (as we have discussed in § 4), we anticipate that this performance superiority is also owing to the `Scaling Tasks per Input` paradigm of our MUFFIN, where the models are trained to focus on the instructions and gain stronger instruction-following capacities. We also find that the larger model benefits more from tuning on MUFFIN (4.42 average performance improvement of T5-3B ⇒ 8.03 average performance improvement of T5-11B, compared with the strongest baseline).

It is noteworthy that nearly all datasets using `Scaling Input-Free Tasks` yield models with limited generalization capabilities towards SUPERNI ( `Scaling-Inputs` ), suggesting challenges for `Scaling Input-Free Tasks` in addressing tasks necessitating supplementary inputs. Furthermore, noting that T0-Eval and BBH belong to `Hybrid` — all the instances from these two benchmarks were initially in line with `Scaling-Inputs` but were transformed into `Scaling Input-Free Tasks` to cater to those `Scaling Input-Free Tasks` datasets. In contrast, our MUFFIN still leads to strong generalization performances, especially on T0-Eval. Considering the tasks with additional input contexts widely exist in real-world applications (e.g., reading comprehension tasks), these results point out the drawbacks in the current `Scaling Input-Free Tasks` paradigm, even if the evaluation tasks have been converted into the paradigm they are familiar with. When considering the comparison with SUPERNI, MUFFIN can still get a comparable or even better performance under some metrics, across different models and model sizes, demonstrating the highly promising instruction-following capacity of the models tuned on MUFFIN.

## 6.2 HUMAN EVALUATION

In this section, we further employ humans to evaluate the quality of different model's responses.

**Acceptance Ratios.** We randomly sample 200 instances from each evaluation benchmark and use various instruction-tuned models to generate the outputs. Then we ask the human evaluator to rate these responses and report the ratio of correct responses from different models. Human evaluation details can be found in Appendix K. Table 16 shows the results. Compared with the strongest results among the 8 baselines, MUFFIN mostly results in better human acceptance ratios with large margins (except for the tiny lower result on T0-Eval). According to our further analysis, we find that the other baselines often misunderstand the task objective, leading to meaningless outputs (e.g., copying some pieces of input text). While MUFFIN is more likely to produce useful outputs to solve the task. We provide some representative cases of different systems' outputs in Table 14.

**Pair-wise Comparison.** We compare the MUFFIN with the strongest baselines by using a pair-wise evaluation scheme, including SELF-INSTRUCT ( `Scaling Input-Free Tasks` ), UNNATURAL INSTRUCT ( `Scaling-Inputs` ) and the human-annotated SUPERNI ( `Scaling-Inputs` ). Similarly, we randomly sample 200 test instances per benchmark and let different models generate the outputs. Table 17 shows the pair-wise comparison results between MUFFIN and other three baselines. Compared with the LLM-generated datasets (SELF-INSTRUCT and UNNATURAL INSTRUCT), our MUFFIN consistently gains higher human preferences across four benchmarks, aligning with the previous conclusion. Notably, owing to the diverse tasks and our novel `Scaling Tasks per Input` paradigm, MUFFIN can even surpass the high-quality human-crafted SUPERNI on three out of four benchmarks.

## 6.3 ANALYSES

In addition to the automatic & human evaluations in the above two subsections, we further conduct some analyses to answer the remaining questions: ($\mathcal{Q}_1$) How much does each technical module (i.e., instruction brainstorm, rematching, classification expansion) contribute to the overall performance? ($\mathcal{Q}_2$) Does our method's superior performance stem from latent task similarities or even leaking towards the evaluation tasks? ($\mathcal{Q}_3$) Does the data size influence the conclusion? ($\mathcal{Q}_4$) Does mixing MUFFIN with human-crafted instruction instances boost the performance?

**Ablation study to answer $\mathcal{Q}_1$.** We fix the data size to 11.5k (because this is the maximum size of rematching instructions; it leads to a fair comparison) and use the combinations of different instruction collecting methods to train the T5-3B. Following Honovich et al. (2022), we report the performances on the validation set of SUPERNI instead of the test set, to avoid cherry-picking. As shown in Table 18, each technical module contributes to overall generalization abilities. Specifically, rematched instructions seem to perform better on classification tasks, while brainstormed instructions demonstrate higher generation performance. Though instruction rematching can gather high-quality instructions written by humans, there is still noise in rematching, and it potentially suffers from imbalanced task categories (e.g., some common tasks are frequently matched with more inputs). Brainstormed instructions are easier to collect, but they are in short supply for classification tasks.

Notably, the proposed classification expansion significantly improves LLMs' generalization on unseen classification tasks while maintaining strong performance on generation tasks.

**Resolving the task leaking concern in $\mathcal{Q}_2$.** It is possible for LLMs to generate the tasks leaking the evaluation benchmarks, violating the zero-shot setting. Therefore, we randomly sample 200 task instructions from each benchmark and use *Sentence Transformers* (Reimers & Gurevych, 2019)[6] to convert all the training instructions and the sampled test instructions into embeddings. We report the average cosine similarity between each training dataset and evaluation benchmark in Figure 6. We also follow the previous works (Yin et al., 2023) using ChatGPT to estimate the task overlap in Table Renze. First, we can find that all the `Scaling-Inputs` datasets, namely SUPERNI, DYNOSAUR and UNNATURAL INSTRUCT, show relatively high instruction semantic similarities with SuperNI-Test (`Scaling-Inputs` as well). Meanwhile, though there are rematched instructions sourced from SUPERNI in our MUFFIN, MUFFIN still maintain a relatively low similarity with SuperNI-Test, owing to the diverse task distribution as discussed in § 4. In a word, our MUFFIN demonstrates relatively low instruction similarities across all four benchmarks, proving the validity of our previous zero-shot instruction-following superiority.

**Scaling data size to answer $\mathcal{Q}_3$.** Regarding the cost efficiency of the previous works, it's usually convenient to collect extensive instruction-following instances with previous paradigms (Yin et al., 2023; Wu et al., 2023). Therefore, *is it worthwhile to try the proposed paradigm rather than simply gathering more instances from the other paradigms?* To answer this question, for all those baseline datasets from the "direct comparison" in Table 2, we randomly sample subsets from them and train T5-3B on the subsets to show the performance trends (10%, 30%, 50%, 80%, 100%). As shown in Figure 9, MUFFIN exceeds the baselines by a noteworthy margin (average scores on four evaluation benchmarks). Other baselines may only be comparable to our data results when they continue to be scaled to several times the size of our data. More importantly, the performances of some datasets even decrease after scaling to a larger size (perhaps due to the noise in these LLM-synthetic datasets), such as SELF-INSTRUCT and DYNOSAUR. Therefore, we conjecture that our paradigm is more efficient than simply collecting more data from the other paradigms. We also conduct the scaling comparison with SUPERNI in Appendix J.1.

**Mixing MUFFIN with SUPERNI to answer $\mathcal{Q}_4$.** As demonstrated by the previous works, it's possible to further improve the generalization performance after mixing the LLM-synthetic and human-annotated instruction instances (Yin et al., 2023). Thus, we mix MUFFIN with the training set of SUPERNI, w.r.t. various proportions, where the proportion means how many instances are from MUFFIN. Then, we train T5-3B on these mixtures and report the average performances on the four benchmarks. As illustrated in Figure 8, interestingly, after mixing MUFFIN and the human-annotated SUPERNI, the performance drops even worse than only using MUFFIN for training. We conjecture possible reasons for this phenomenon: *the dataset paradigm does affect the learning efficiency* — different dataset paradigms do obviously have various impacts on the model's performance. Therefore, combining two paradigms can even hurt the model's instruction-following performance, meaning the paradigm is critical. Meanwhile, the effect of dataset paradigms is even greater than that of data quality (i.e., adding a small proportion of high-quality data from other paradigms even harms MUFFIN's performance), further implying the proposed paradigm's effectiveness.

## 7 CONCLUSION

This work proposes a novel scheme for curating instruction-following datasets, namely `Scaling Tasks per Input`. Unlike previous works that either adopt `Scaling-Inputs` or `Scaling Input-Free Tasks` paradigm, we diversify the tasks for each input text. The variance in task instructions leading to different outputs can ideally enhance the instruction-following capacity of LMs. Accordingly, we propose MUFFIN — the first dataset aligning with `Scaling Tasks per Input`. Our comprehensive experiments spanning four challenging zero-shot benchmarks demonstrate the effectiveness of MUFFIN, where MUFFIN consistently achieves better instruction-following capacity than the extensive baselines with previous paradigms. In-depth human evaluation and analyses further prove the superiority of MUFFIN and our `Scaling Tasks per Input` paradigm.

---

[6]https://huggingface.co/sentence-transformers/all-MiniLM-L6-v2/tree/main

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

APPENDICES

Within this supplementary material, we elaborate on the following aspects:

- Appendix A: Details on Sampling Inputs and Instructions
- Appendix B: Details on Instruction Rematching
- Appendix C: Data Collection Prompt Templates and Hyper-parameters
- Appendix D: Data Statistics
- Appendix E: Data Diversity
- Appendix F: Data Collection Cost and API Usage
- Appendix G: Details of Tuning MUFFIN
- Appendix H: Details of Baselines
- Appendix I: Training and Evaluation Prompts
- Appendix J: Further Analyses
- Appendix K: Human Evaluation Details

## A DETAILS ON SAMPLING INPUTS AND INSTRUCTIONS

As mentioned in previous sections, we use both inputs and instructions from SUPERNI to construct our MUFFIN. In this section, we elaborate on the sampling details.

When sampling inputs (§ 3.1), since the construction of SUPERNI is based on the existing NLP benchmarks, different tasks in SUPERNI may source from the same benchmark. Therefore, in order to promote the diversity of input texts, we only sample inputs of tasks from unique sources (a total of 243 out of 756 tasks from SUPERNI's training set). For each resulting task, we randomly pick up 4 input texts. We conduct some instruction filtering (§ 3.2) that helps delete the non-answerable instructions, leading to a final of 953 inputs from SUPERNI have valid instruction and output annotations (see Table 1).

As for the choices of the instructions used in "*instruction rematching*" (§ 3.2), we use all the instructions in the 757 training tasks of SUPERNI. Since we also use the task inputs from SUPERNI, those inputs and instructions that are matched in the origin SUPERNI will be automatically matched together in our dataset as well. However, due to using inputs from unique sources, there are a certain amount of instructions that aren't matched with any inputs, resulting in a total of 574 instructions in our dataset sourced from SUPERNI (as shown in Table 1).

## B DETAILS ON INSTRUCTION REMATCHING

As introduced in § 3.2, we use the LLM, namely GPT-4, to help gather existing human-written instructions for each input. However, classifying such a large number of candidate (input, instruction) pairs, where the majority are negative, is pretty costly and inefficient. To solve this issue, inspired by the "*entailment check*" of Xie et al. (2023); Gu et al. (2023), we first adopt a free and small LM (SLM), Flan-T5 (Chung et al., 2022),[7] for quick and rough filtering to reduce the number of candidate pairs, then call the LLM to further filter on the kept pairs. Moreover, those pairs annotated by LLM can be further used for training the SLM, improving the subsequent rough filtering quality. This SLM-LLM collaboration significantly improves the annotation efficiency. According to our small-scale trials, we found that this method can reduce the annotation cost to about 35% of the direct annotation of LLM.

## C DATA COLLECTION PROMPT TEMPLATES AND HYPER-PARAMETERS

**Facet Recognition**    We show the prompt used for generating facets (also can be understood as "attributes") in Table 3. As for the decoding parameters of OpenAI API, we set the temperature as

---

[7]https://huggingface.co/google/flan-t5-large/tree/main

Table 3: Prompt used in *Facet (Attribute) Recognition* § 3.2.

```
### Input:
{Input Text}

### Instruction:
Given the above input, what kind of textual attributes does it have?

### Requirements:
1. Please brainstorm as many textual attributes as possible. If you think there are no more suitable attributes, end up with 'None'.
2. Be creative. Any interesting perspectives are welcome!
3. Each attribute must concisely summarize one specific aspect of this input, such as language, length, intent, etc.
4. Feel free to ignore the tedious and specific content. Just focus on some general textual attributes!
5. Please prioritize your most confident predictions.

Attribute 1:
```

0.7, and nucleus sampling (top p) as 1.0 to promote the diversity of attributes. As for the maximum generation tokens, we fix it as 2,048.

**Instruction Brainstorm**   Table 4 shows the prompts used in generating instructions. When running instruction brainstorm, we fix the temperature to 0.2 and set the nucleus sampling (top p) as p = 0.99. We set the maximum generation tokens as 3,200 to let LLMs brainstorm more complex instructions. We also set an additional presence penalty parameter as 1.99 to encourage LLMs to produce more diverse tasks.

**Instruction Rematching**   Table 5 illustrated the prompt used in instruction rematching, namely letting LLMs decide whether an instruction is appropriate for the given input. We set the temperature as 0.2 and the top p as 0.0 in this case to make the decision more deterministic. As for the maximum generation tokens, we set it to 2,048.

**Output Annotation**   Table 6 shows the prompt when generating the outputs for a given (instruction, input) pair. To ensure the determinism of outputs, we set the temperature as 0.1 and the top p as 0.1 as well. As for the maximum generation tokens, we set it to 1,024.

**Classification Expansion**   We illustrated the prompt of classification expansion in Table 7. When asking the LLMs to generate wrong answer candidates, we empirically set the temperature as 0.3 and the top p as 0.99. As for the maximum generation tokens, we set it to 3,096.

Table 5: Prompt used in *Instruction Rematching* § 3.2.

```
You are an expert in Natural Language Processing (NLP) tasks.
Given a task description and a piece of text, your job is to determine whether
this text can be used as input for this task.
If the text satisfies the input expectation of this task, answer 'Yes'; otherwise,
if the text doesn't match the input description, answer 'No'.

### Text:
{Input Text}

### Task Description:
{Task Instruction}

### Your Answer:
```

Table 4: Prompt used in *Instruction Brainstorm* § 3.2. We actually use two different prompts in our experiments: 1) the first prompt (left table) uses the textual attribute as the *hint* and asks the LLMs to follow the hint to generate the corresponding instructions (Li et al., 2023); 2) in contrast, the second prompt (right table) asks the LLMs to develop task instructions that try to *shift* the given attribute. For example, if the attribute is something like "`the input is lengthy`", then we hope the LLMs can brainstorm some tasks like "`summarize and simplify the given input`". In our preliminary experiments, we found that the two different prompt strategies can result in complementary task categories. Therefore, for each input, we collect the task instructions produced by using both prompts. **The differences between these two prompts are highlighted.**

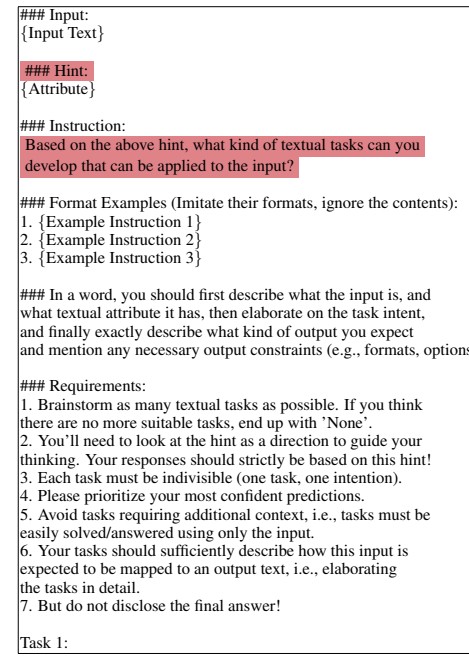
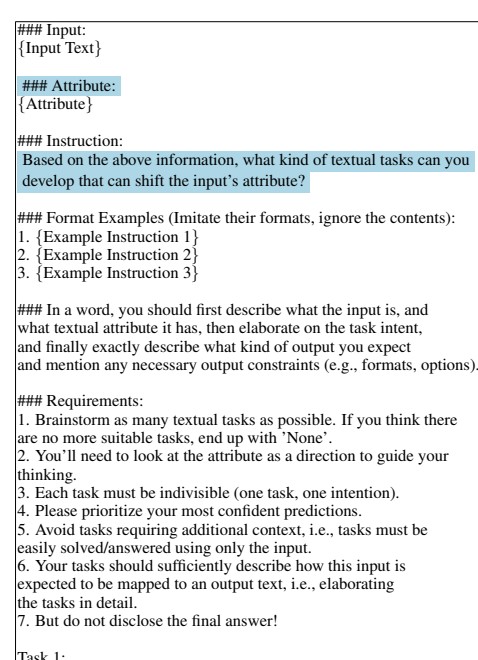

Table 6: Prompt used in *Output Annotation* § 3.2.

Given a task instruction and an input, please generate the output (answer) according to
the requirements mentioned in the instruction.
If you cannot answer the instruction base on the given information, simply generate 'None'.

### Instruction:
{Task instruction}

### Input:
{Input Text}

### Output:

Table 7: Prompt used in *Classification Expansion* § 3.3. We ask the LLMs to generate more output
candidates that are worse than the given output, which can be further used for reformulating the
origin task into a classification paradigm.

Given a task, a task input, and a corresponding correct output, generate more output
candidates for this task.

### Requirements:
1. The output candidates you generate should be worse than the given correct output,
e.g., wrong or imperfect answers.
2. You are encouraged to generate some challenging output candidates, that are
close to the correct output but not the most desired one (i.e., containing certain errors).
3. You are encouraged to generate as many output candidates as possible; If you think
there are no more suitable output candidates, end up with 'None'.

### Task:
{Task Instruction}

### Input:
{Input Text}

### Output:
{Output Text}

Wrong Output 1:

# D   DATA STATISTICS

We report the length distribution of the inputs, instructions, and outputs of MUFFIN in Figure 4. All the length is counted by words (we follow previous works using NLTK package[8] to conduct the word tokenization). Our dataset covers a diverse length distribution (see the maximum and minimum length of each figure), however, the most common length is still highly focused in a certain range, as the average length reported in Table 1.

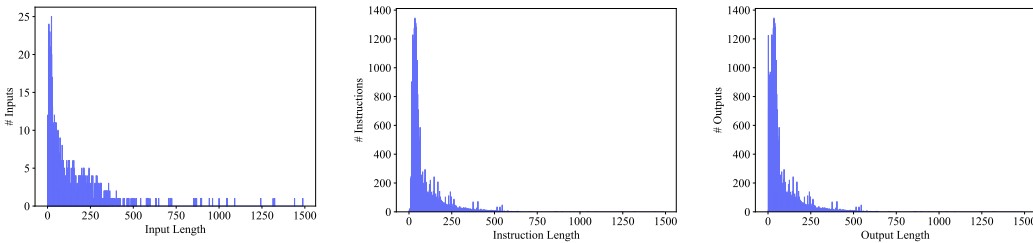

Figure 4: Length distribution of inputs, instructions, and outputs in our MUFFIN.

# E   DATA DIVERSITY

Similar to Zhang et al. (2023a), we use Berkeley Neural Parser[9] to phrase the verb-noun structure of the instructions in MUFFIN. Figure 5 shows the most common verb-noun structures of our dataset, which implies a high task diversity of MUFFIN.

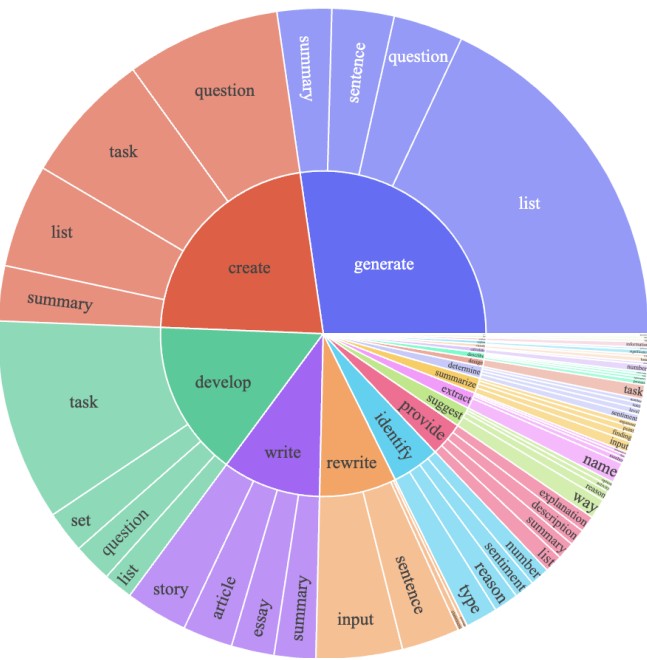

Figure 5: Instruction diversity illustration in § 4. We plot the top 20 most prevalent root verbs (inner circle) and their top 4 direct nouns (outer circle) in the instructions of MUFFIN.

---

[8] https://pypi.org/project/nltk/
[9] https://parser.kitaev.io/

## F DATA COLLECTION COST AND API USAGE

During our preliminary trials, we observed GPT-4's superiority over ChatGPT in instruction rematching (§ 3.2), which can produce much fewer false-positive pairs. Therefore, in our dataset collection pipeline, we mainly use two LLMs: 1) ChatGPT for instruction brainstorming and 2) GPT-4 for instruction rematching. Table 8 provides the detailed API usage and the corresponding cost of each phase in our pipeline. As for the reason for using different APIs of ChatGPT: we found that the `gpt-3.5-turbo-0301` demonstrates better creativity than `gpt-3.5-turbo-0613`, where `gpt-3.5-turbo-0301` constantly brainstorms more task instructions. In contrast, `gpt-3.5-turbo-0613` tends to be more accurate on annotations than `gpt-3.5-turbo-0301`. Thus, we use `gpt-3.5-turbo-0301` for generating diverse textual attributes and instructions, while using `gpt-3.5-turbo-0613` to produce the outputs.

Table 8: The overall cost and API usage.

|  | API | Cost |
|---|---|---|
| **Brainstormed instructions (56,495)** |  | **$182.90** |
| - Facets recognition | gpt-3.5-turbo-0301 | $5.49 |
| - Instruction brainstorm | gpt-3.5-turbo-0301 | $49.11 |
| - Output annotation | gpt-3.5-turbo-0613 | $58.56 |
| - Classification expansion | gpt-3.5-turbo-0613 | $69.74 |
| **SuperNI rematching instructions (11,519)** |  | **$388.44** |
| - Instruction-input rematching | gpt-4-0613 | $271.08 |
| - Output annotation | gpt-4-0613 | $117.36 |
| **Total (68,014)** |  | **$571.34** |

## G DETAILS OF TUNING MUFFIN

All of our implementations are based on HuggingFace transformers (Wolf et al., 2019).[10] For the implementation of tuning T5 models, we adopt the source training code of SUPERNI;[11] for fine-tuning the Llama2, we use the open-sourced Alpaca-Lora,[12] where we apply LoRA (Hu et al., 2022) to relieve the high computational cost. Besides, all the models' weights are downloaded from HuggingFace repositories.[13] We fine-tune T5 on MUFFIN with 2 epochs. When fine-tuning T5-3B, we set the learning rate as $5e-5$ with batch size 6. As for T5-11B, we set the learning rate as $1e-5$ with batch size 1 and 12 gradient accumulation steps. All the above hyper-parameters are tuned on the validation set of SUPERNI. While we fine-tune Llama2 on all the datasets 3 epochs with batch size 18, and we set learning rate $= 1e-4$, $lora_r = 8$, $lora_{alpha} = 16$. Since the generation API provided by HugggingFace cannot support efficient batched evaluation, we fix the evaluation batch size to 1 for all the datasets. We truncate the inputs to 1024 tokens and limit the output length to 128 tokens, with beam search size $= 1$ (greedy decoding).

All the experiments are done on NVIDIA A100 with 80 GPU memories. When fine-tuning T5 models, we utilize DeepSpeed ZeRO stage 2 and 3 for single-GPU and multiple-GPU tuning, respectively.

## H DETAILS OF BASELINES

We elaborate on more details of each baseline dataset used in our experiments. Following Honovich et al. (2022), all the hyper-parameters of fine-tuning baselines are trialed on the validation set of SUPERNI. We truncate all the input length into 1024 and the output length into 128 (except those datasets designed for long response generation), which is the same as fine-tuning our MUFFIN. For a

---

[10]https://pypi.org/project/transformers/

[11]https://github.com/yizhongw/Tk-Instruct

[12]https://github.com/tloen/alpaca-lora

[13]We use T5-3B from https://huggingface.co/t5-3b/tree/main, T5-11B from https://huggingface.co/t5-11b/tree/main, and Llama2-13B from https://huggingface.co/meta-llama.

fair comparison, analogous to (Yin et al., 2023), we keep similar data size for all systems (60k $\sim$ 80k) with downsampling when necessary, and report the average performance of three seeds. When evaluating the models, we use the official scripts (Rajpurkar et al., 2016) [14] to calculate the metrics, which are widely adopted by the previous works.

**Dolly (Conover et al., 2023)** `Scaling Input-Free Tasks` is a human-annotated instruction dataset created by following the schemes of InstructGPT (Ouyang et al., 2022) and open-ended free-form paradigm. We use the officially released dataset of DOLLY.[15] We fix the learning rate as `5e-05` and `3e-05` for T5-3B and T5-11B, respectively. We fine-tune models on DOLLY with 2 epochs.

**LongForm (Köksal et al., 2023)** `Scaling Input-Free Tasks` is a LLM-generated dataset that is specifically proposed for enhancing long text generation.[16] Due to the lengthy output of LONGFORM, we extend its output length limitation into 512 when fine-tuning. We fine-tune models onLONGFORM with 2 epochs. We set the learning rate of T5-3B as `3e-5` and T5-11B as `2e-5`.

**Alpaca (Taori et al., 2023)** `Scaling Input-Free Tasks` is an LLM-generated dataset that mainly follows the same data construction pipeline as SELF-INSTRUCT. The main difference of ALPACA is using a more powerful API. We use the official version of ALPACA.[17] We fine-tune models on ALPACA with 3 epochs. We set the learning rate of T5-3B and T5-11B as `5e-5`.

**Alpaca-GPT4 (Peng et al., 2023)** `Scaling Input-Free Tasks` is similar to ALPACA but uses GPT-4 as the API to synthesize data.[18] We fine-tune models on ALPACA-GPT4 with 3 epochs. We set the learning rate of T5-3B and T5-11B as `5e-5`.

**WizardLM (Xu et al., 2023a)** `Scaling Input-Free Tasks` uses LLMs to evolve existing instructions (e.g., the seed instructions from ALPACA) into a more complicated version. At the time of writing, we use the latest version of WIZARDLM for experiments.[19] similar to LONGFORM, the outputs in WIZARDLM dataset are lengthy. Therefore, we set the output length as 512. We fine-tune models with 2 epochs and fix the learning rate of T5 (3B and 11B) to `5e-5`.

**Self-Instruct (Wang et al., 2023b)** `Scaling Input-Free Tasks` uses existing human-crafted user instructions as seeds and let GPT-3 (Brown et al., 2020) devise novel task instructions.[20] We fine-tune models on SELF-INSTRUCT with 3 epochs. We fix the learning rate of T5-3B and T5-11B to `5e-5` and `2e-5`, respectively.

**Unnatural Instruct (Honovich et al., 2022)** `Scaling-Inputs` There are two versions of UNNATURAL INSTRUCT, namely the "core" version and the "paraphrase-expanded" version. Since the paraphrase-expanded version is specifically designed for fitting T0-Eval and BBH evaluation benchmarks, we use the core set of UNNATURAL INSTRUCT in our experiments for a fair comparison.[21] We fine-tune models on UNNATURAL INSTRUCT with 3 epochs. As for the learning rate, we set `5e-5` and `1e-5` for T5-3B and T5-11B, respectively.

**Dynosaur (Yin et al., 2023)** `Scaling-Inputs` utilizes the input-output pairs in existing NLP datasets and drives LLMs to recover the potential tasks for each dataset, significantly reducing the data synthesis cost. We use the officially released version of DYNOSAUR,[22] where Yin et al. (2023) used ChatGPT to filter those task categories that are similar to the tasks in SuperNI-Test. We set the epoch as 2 and fine-tune both T5-3B and T5-11B with a learning rate of `1e-5`.

---

[14] `https://github.com/yizhongw/Tk-Instruct/blob/main/src/compute_metrics.py`

[15] `https://huggingface.co/datasets/databricks/databricks-dolly-15k`

[16] `https://github.com/akoksal/LongForm/tree/main/dataset`

[17] `https://github.com/tatsu-lab/stanford_alpaca`

[18] `https://github.com/Instruction-Tuning-with-GPT-4/GPT-4-LLM`

[19] `https://huggingface.co/datasets/WizardLM/WizardLM_evol_instruct_V2_196k`

[20] `https://github.com/yizhongw/self-instruct/tree/main/data`

[21] `https://github.com/orhonovich/unnatural-instructions/blob/main/data/core_data.zip`

[22] `https://huggingface.co/datasets/Dynosaur/dynosaur-sub-superni`

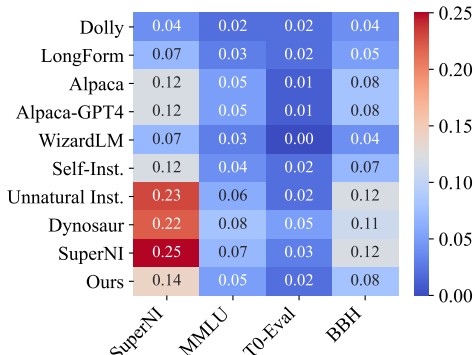

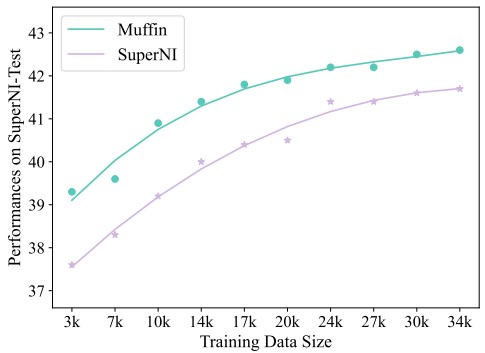

Figure 6: The instruction semantic similarity between each training dataset and evaluation benchmark.

Figure 7: The scaling trends comparison between SUPERNI and our MUFFIN. The performances are based on T5-3B.

**SuperNI (Wang et al., 2022)** `Scaling-Inputs` is human-crafted high-quality instruction dataset, covering more than 1,600 tasks either from existing NLP benchmarks or created by human expertise.[23] When training on SUPERNI, we follow the official implementation tunning models on the 756 training tasks with a sequential multi-task learning scheme. We use the official code and hyper-parameter settings of (Wang et al., 2022) to reproduce the performances.

## I  TRAINING AND EVALUATION PROMPTS

During the training and evaluation, we use the similar prompt template as the implementation of Wang et al. (2022),[24] where we concatenate the task instruction and input together and ask the model to produce the output for this instance. To be specific, we use the following prompt: "`### Input:\n{Input Text}\n\n.### Instruction:{Task Instruction}\n\n### Output:`". We fix this prompt for all the methods in our experiments (including our MUFFIN and all the baselines).

It is worth noting that, there are instances in some baseline datasets that don't have the task input, such as SELF-INSTRUCT and ALPACA. As for those dataset, we set the "{Input Text}" filed as "`None`". Similarly, the instances in evaluation benchmarks, namely MMLU, T0-Eval, and BBH, also have empty task inputs, we set them to "`None`" as well. This also implies that, compared with those input-free datasets, our MUFFIN is more disadvantageous in the generalization of these benchmark datasets. However, MUFFIN can still yield better generalization capacity according to the experiments in § 6.

## J  FURTHER ANALYSES

### J.1  SCALING TRENDS COMPARISON WITH SUPERNI

As we introduced in § 1, the `Scaling-Inputs` paradigm scales the input-output pairs for each task, while our paradigm scales tasks for each input text. These two paradigms follow totally opposite scaling philosophies. Therefore, it's interesting to compare the scaling efficiency between them. To this end, we compare the generalization performances of SUPERNI and MUFFIN. For SUPERNI, we fix the task number as 681 (the maximum training task size as we mentioned in § 5), and scale *inputs per task*; similarly, for MUFFIN, we fix the input number as 681 and scale the *tasks per input*. We range both *inputs per task* and *tasks per input* from 5 to 50, as the maximum inputs per instruction of MUFFIN is 46.68 (see Table 1). We train T5-3B and report the overall *ROUGE-L* performances on the SuperNI-Test. Since SUPERNI is a human-annotated dataset, we use ChatGPT to reannotate the outputs of SUPERNI (using the same prompt and setting as our output annotation, see Table 6) to

---

[23]https://instructions.apps.allenai.org/
[24]https://github.com/yizhongw/Tk-Instruct

Table 9: The task category overlap between different training datasets and evaluation benchmarks, estimated by ChatGPT.

|  | SuperNI-Test | MMLU | T0-Eval | BBH |
|---|---|---|---|---|
| Self-Inst. | 11.62% | 10.37% | 11.27% | 5.29% |
| Unnatural Inst. | 22.27% | 16.91% | 18.39% | 4.76% |
| Dynosaur | 12.08% | 13.49% | 15.20% | 1.27% |
| Muffin | 16.01% | 13.05% | 8.63% | 2.53% |

Table 10: Create MUFFIN with the input sourced from DYNOSAUR and UNNATURAL INSTRUCT (namely MUFFIN-DYNOSAUR and MUFFIN-UNNATURAL). All the datasets here have around 16k instances. The experiment is based on T5-3B.

| Models | SupnerNI-Test (overall) | MMLU (ACC) | MMLU (EM) | T0-Eval (ACC) | T0-Eval (EM) | BBH (EM) |
|---|---|---|---|---|---|---|
| Unnatural Inst. | 37.12 | 24.35 | 22.33 | 45.93 | 35.88 | 8.69 |
| Dynosaur | 32.55 | 26.88 | 25.26 | 38.56 | 39.13 | 12.09 |
| Muffin-Unnatural | 38.86 | 33.12 | 24.73 | **46.08** | 43.63 | 12.91 |
| Muffin-Dynosaur | 38.23 | **33.34** | 22.59 | 45.3 | **44.01** | **13.53** |
| Muffin | **40.45** | 33.24 | **25.39** | 42.68 | 40.99 | 13.44 |

ensure a more fair comparison. Figure 7 shows the results. Observably, our MUFFIN consistently demonstrates a better generalization capacity compared with SUPERNI. Note that, even though the outputs of SUPERNI are reannotated, the relatedness of (instruction, input) pairs in SUPERNI is still better than MUFFIN (refer to Figure 3). Therefore, it further proves the superiority of MUFFIN and suggests the effectiveness of the proposed `Scaling Tasks per Input` paradigm.

## J.2 THE TASK OVERLAP ESTIMATION

Beyond the task similarity analysis provided in § 6.3, we follow previous work prompting ChatGPT to further quantify the task category overlap between the evaluation benchmarks and different training datasets (Yin et al., 2023). Given an instance from a training dataset, we ask ChatGPT whether it belongs to any task categories from the evaluation benchmark. For example, the test set of SUPERNI has 12 task categories, and we request ChatGPT to perform a 13-way classification (one "None of above" label) on the training instances. Finally, we report how many instances from this training dataset contain task leakage.

Table 9 illustrates the results. Here, we only report the overlap results of MUFFIN and those competitive baseline datasets according to Table 2. Though there are some shifts in the leakage ranking compared with the semantic-based analysis in § 6.3 (see Figure 6), the overall conclusion is still the same — MUFFIN demonstrates relatively low evaluation task leakage across all four benchmarks.

## J.3 USING THE INPUTS SOURCED FROM THE OTHER LLM-SYNTHETIC DATASETS

As introduced in § 3, the creation of MUFFIN utilizes the human-crafted inputs from SUPERNI (i.e., "instruction brainstorm"). While those baselines with `Scaling-Inputs` paradigm either utilize inputs from other human-crafted datasets (e.g., DYNOSAUR) or let LLMs to synthesize inputs. To investigate whether the proposed "instruction brainstorm" method can be extended to other input sources, we gather the inputs from other LLM-generated datasets (including DYNOSAUR and UNNATURAL INSTRUCT), and apply our "insruction brainstorm" method to them. Finally, we create two new small-scale datasets (around 16k instances), namely MUFFIN-DYNOSAUR and MUFFIN-UNNATURAL, and compare their performances with DYNOSAUR and UNNATURAL INSTRUCT (in the same size). Table 10 shows the results, where we observe that MUFFIN consistently demonstrates superior generalization performance, regardless of the text sources employed. It further suggests that MUFFIN's robustness to follow instructions is not reliant on the input resources but rather stems more from our crucial contribution—the diversified instructions per input. Therefore, we anticipate that our method can be ideally applied to any input resources while still crafting diverse task instructions.

Table 11: The statistics of generated facets.

| facets statistics | |
|---|---|
| # of inputs | 1,463 |
| # of instructions (from facets-based brainstorm) | 33,720 |
| # of facets | 11,382 |
| avg # of facets per input | 7.78 |
| max # of facets per input | 30 |
| min # of facets per input | 2 |
| avg # of instructions per facet | 2.96 |
| max # of instructions per facet | 18 |
| min # of instructions per facet | 1 |

Table 12: The quality evaluation results of generated facets.

| | |
|---|---|
| input-to-facet correctness | 90.78% |
| facet-to-instruction correctness | 85.22% |

## J.4 THE EXPLANABILITY OF THE GENERATED FACETS

As mentioned in § 3, the proposed "instruction brainstorm" method drives LLMs to brainstorm novel instructions based on the synthetic facets. To this end, it's unclear whether the synthetic facets are explainable and reliable for the subsequent instruction generation due to the LLM's hallucinations (Xie et al., 2023; Zhang et al., 2023b). Therefore, in this section we conduct several additional analysis on the generated facets in MUFFIN, including **statistics**, **quality**, and **diversity** of the facets.

**Facets Statistics.** Beyond the statistics of instructions in Table 1, we report additional statistics related to facets in Table 11 (the statistics after conducting "instruction filtering").

**Factes Quality.** To estimate the quality and reliability of the generated facets, we conduct a further human verification on the facets. Specifically, we randomly collect 100 inputs from MUFFIN, along with their corresponding facets (889 facets in total) and instructions (2,265 instructions in total). Then, we ask an experienced human annotator to evaluate the following two metrics:

- **Input-to-facet correctness**: whether each facet correctly described the given input (reflecting the quality and reliability of facts).

- **Facet-to-instruction correctness**: whether the subsequent instructions are reasonably related to the given facet (reflecting the utility of facets).

The final results are shown in Table 12. These results demonstrate the good quality and utility of the facets generated by our method. It's also worth noting that, after we conducted "instruction filtering" (as introduced on § 3), the input-to-facets correctness increased from 83.28% to 90.78%, proving the effectiveness of the naive filtering method that can jointly filter a considerable amount of hallucinated facets.

**Facets Diversity.** As for the diversity of the generated facets, we ask the human annotator to *discover* the topic categories from the facets (i.e., clustering instead of classification), because it's hard to predefine the categories that may introduce personal bias. We used the same instances as in the previous analysis (100 inputs and 889 facets). The human annotator is asked to summarize a "keyword" (topic category) for each LLM-generated facet. If the current category has already appeared before, the annotator also has to "cluster" those facets that belong to the same category together. In doing so, more and more novel categories can be discovered.

We then calculate the following two metrics:

- **Intra-diversity**: how many unique categories there were for each input's facets, and reported an averaged ratio among all the inputs.

- **Inter-diversity**: overall categories distribution among all the 889 facets, namely the global frequency of each unique facet.

We find that the generated facets achieve 91.55% intra-diversity, meaning the facets of the same input are pretty diverse. Meanwhile, there are a total of 276 unique categories out of the 889 facets, which implies a high inter-diversity of the facets.

Table 13: Example cases in our dataset, including both high-quality and low-quality cases. We employ human annotators to judge the validity of instances. As mentioned in § 4, we develop three validity questions (binary judgments). "Instruction" means "Q1: Does the instruction describe a valid task that can be answered?"; "Input" means "Q2: Does this instruction appropriately match the input?"; while "Output" means "Q3: Does the output correctly respond to the instruction and input?"

| Task Instruction | Task Input | Task Output | Validity |
|---|---|---|---|
| In this task, you are given a sentence that describes the defeat of one superhero by another. Your task is to generate a list of adjectives that describe the defeated superhero's state after their defeat. The output should consist of at least three adjectives separated by commas. | ISuperior Spider-Man appears in LEGO Marvel Super Heroes and is quickly dispatched by the original Spidey. | defeated, humiliated, weakened | Instruction: ✓ Input: ✓ Output: ✓ |
| Given a sentence, generate what should be the most likely next statement. The next statement should be reasonable and logically correct. | Would you like to go shopping together this Saturday? | Sure, I would love to go shopping with you this Saturday! | Instruction: ✓ Input: ✓ Output: ✓ |
| Write a news article reporting on an incident where someone was found with a numb expression. Include quotes from witnesses and authorities, as well as any relevant information about the incident. | He continues out with a numb expression. | Title: Mysterious Incident Leaves Man with Numb Expression Date: [*Current Date*] In a bizarre turn of events, a man was found with a numb expression in an incident that has left authorities puzzled. The incident occurred earlier today in [*Location*], and witnesses [...] | Instruction: ✓ Input: ✓ Output: ✓ |
| You are given a set of three numerical values. Your task is to determine the average value of these numbers and return it as a float with two decimal places. | [87.487, 27.095, 27.812] | 47.13 | Instruction: ✓ Input: ✓ Output: ✗ |
| In this task, you're given a sentence and your task is to generate a simplified sentence that focuses on lexical paraphrasing from the original sentence given in the input. | Text: Cymande is the self-titled debut album by British funk group Cymande released in 1972. Title: Cymande (album) | The self-titled debut album by British funk group Cymande was released in 1972. | Instruction: ✓ Input: ✓ Output: ✗ |
| In this task, you are given a set of mathematical expressions. Your task is to identify the expression that has the highest value. Output: The mathematical expression with the highest value. | What is (-1 - 0) + -5 + -6 + 8? | 73 | Instruction: ✓ Input: ✗ Output: ✗ |
| Given the input, generate a list of potential reasons why there are zero comments. Each reason should be a complete sentence and should not repeat any information already given in the input. | topic: InfraRedEd sentence: Zero comments. | There are no comments because the topic is not popular. | Instruction: ✓ Input: ✗ Output: ✗ |
| Develop a task to extract the names of the actors who played the characters David, Monica, and Gigolo Joe in the movie "A.I." from the given review. The output should be a list of these three actor names. | it is easy to see why the late stanley kubrick believed steven spielberg would be the perfect director for his artificial intelligence, just as it is understandable why spielberg was attracted to the material. | Haley Joel Osment, Frances O'Connor, Jude Law | Instruction: ✗ Input: ✗ Output: ✗ |
| In this task, you are given a description of how Lassi is traditionally made. Your job is to extract all the steps involved in making it and return them as a numbered list. | Lassi. | 1. Blend yogurt, water, and sugar together. 2. Add crushed ice and blend again until smooth [...] | Instruction: ✗ Input: ✗ Output: ✗ |

## J.5 DATA CASES WITH HUMAN-EVALUATION VALIDITY

We provide some representative cases (instructions, inputs, outputs) of our dataset in Table 13, including both high-quality cases (all three fields are correct) and low-quality cases (some of the fields are incorrect).

## J.6 SYSTEM OUTPUT CASES ON THE EVALUATION BENCHMARKS

Table 14 reports the system outputs of LMs tuned on different datasets (baselines and MUFFIN).

Table 14: Output cases of baselines and MUFFIN across four benchmarks. "Self-Inst." and "Unnatural" indicate the SELF-INSTRUCT and UNNATURAL INSTRUCT, respectively.

| Category | SuperNI-Test | | |
|---|---|---|---|
| | Answer Generation | Question Modification | Summarization |
| **Instruction** | Given an abstract, and a few options, indicate if any of the given options appropriately describe the given abstract. If any of them is a proper keyword, answer 'Yes', otherwise answer 'No'. | You're given a fill-in-the-blank question where the answer is PersonX. You need to minimally change the given question [...] than PersonY. Although there are many correct answers, you only need to write one of them. | Generate a topic word for the given sentence. A topic word is usually the name of a person, place, animal, concept or object the sentence is focusing on. |
| **Input** | Abstract: A behavioural method of habit reversal, [...]. Question: Does the following mesh [...] Support, Non-U.S. Gov't. | Context Word: summer. Question: The summer went too quickly for PersonX. PersonY was excited for school, but _ didn't want to go back. Answer: PersonX | the chinese postman problem is a mathematical problem of graph theory. |
| **Ground Truth** | Yes | The summer went too quickly for PersonX. PersonY was excited for school, and _ wanted to go back. | chinese postman problem |
| **Self-Inst.** | No | PersonY | Chinese |
| **Unnatural** | 3)Behavior Therapy | PersonY was excited for school | graph theory |
| **Muffin (Ours)** | Yes | The summer went too quickly for PersonY. PersonY was excited for school, but _ didn't want to go back. | chinese postman |

| Category | MMLU | | |
|---|---|---|---|
| | Logic Test | Sexuality Test | Fact Test |
| **Instruction** | Use indirect truth tables to determine whether the following argument is valid. [...] (A): Valid (B): Invalid. Counterexample [...] | What makes a relationships last, according to Sternberg? (A): intimacy (B): commitment (C): decision (D): passion | As of 2019, about what percentage of people from Turkey agree that the state is run for the benefit of all the people? (A): 36% (B): 51% (C): 66% (D): 81% |
| **Input** | / | / | / |
| **Ground Truth** | A | B | C |
| **Self-Inst.** | (A): Valid (B): Invalid (C): Invalid (D): Invalid | (D) | (A) |
| **Unnatural** | B | (B) | A: 36% B: 51% C: 66% D: 81% |
| **Muffin (Ours)** | A | (B) commitment | (C): 66% |

| Category | T0-Eval | | |
|---|---|---|---|
| | Causal Reasoning | Natural Language Inference | Natural Language Inference |
| **Instruction** | The man broke his toe. I am hesitating between two options. Help me choose the more likely cause: [...] | Given that Hey, look who's here [...] Does it follow that Annunaki is retiring after this episode. Yes, no, or maybe? | Suppose it's true that "Fight or Flight" is the fifth episode of the second season of the NBC science fiction drama series [...] always, sometimes, or never true? |
| **Input** | / | / | / |
| **Ground Truth** | He dropped a hammer on his foot. | No | Sometimes |
| **Self-Inst.** | (a) | Yes | The episode aired at 8 PM est. is sometimes true. |
| **Unnatural** | He got a hole in his sock | Yes, no | TRUE |
| **Muffin (Ours)** | _ He dropped a hammer on his foot. | No | Sometimes |

| Category | BBH | | |
|---|---|---|---|
| | Error Detection | Word Sorting | Object Counting |
| **Instruction** | The following translations from German to English contain a particular error. [...] Options: (A) Modifiers or Adjectives (B) Numerical Values (C) Negation or Antonyms (D) Named Entities (E) Dropped Content (F) Facts | Sort the following words alphabetically: List: alleviate duopoly mattress gland benelux townsmen buoyant klaxon hardbound tomography felice gunk | I have a mouse, a rabbit, a dog, a duck, and two goats. How many animals do I have? |
| **Input** | / | / | / |
| **Ground Truth** | (A) | alleviate benelux buoyant duopoly felice gland gunk hardbound klaxon mattress tomography townsmen | 6 |
| **Self-Inst.** | D) Named Entities | ['a', 'e', 'i', 'o', 'u'] | I have 6 animals. |
| **Unnatural** | (F) | List: alleviate duopoly mattress gland benelux townsmen buoyant | I have six animals. |
| **Muffin (Ours)** | (A) Modifiers or Adjectives | alleviate benelux buoyant duopoly felice gland gunk hardbound klaxon mattress tomography townsmen | 6 |

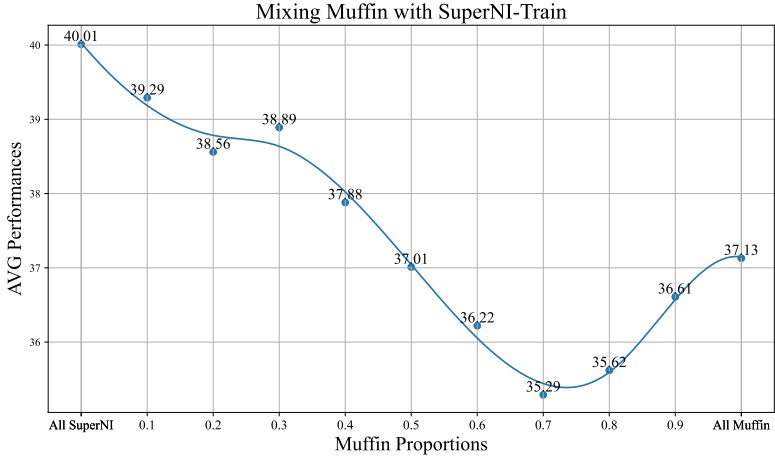

Figure 8: The performance of the mixture of MUFFIN and SUPERNI.

Table 15: Results by Llama2-13B (Touvron et al., 2023b). For the sake of simplicity, we omit the *Rank Classification Accuracy* here. The best scores under direct and indirect comparison settings are in **bold** and underlined, respectively.

| Models | Data Size | SuperNI-Test | | | MMLU | T0-Eval | BBH |
|---|---|---|---|---|---|---|---|
| | | *EM (CLS)* | *ROUGE-L (GEN)* | *ROUGE-L (overall)* | *EM* | *EM* | *EM* |
| *Human Annotated Data (indirect comparison)* | | | | | | | |
| SuperNI | 68k | 50.73 | 55.99 | 52.43 | 31.38 | 46.37 | 12.26 |
| *Generated Data (direct comprison)* | | | | | | | |
| Dolly | 15k | 9.96 | 43.58 | 27.25 | 0.39 | 22.29 | 7.76 |
| LongForm | 23k | 4.30 | 41.30 | 19.07 | 0.12 | 0.72 | 5.27 |
| Alpaca | 52k | 33.34 | 51.67 | 43.65 | 36.01 | 40.39 | 21.72 |
| Alpaca-GPT4 | 52k | 18.27 | 44.27 | 33.50 | 1.01 | 6.29 | 2.20 |
| WizardLM | 68k | 10.52 | 43.36 | 27.27 | 0.29 | 7.20 | 4.24 |
| Self-Inst. | 82k | 36.82 | 46.79 | 41.04 | 23.12 | 31.43 | **28.69** |
| Unnatural Inst. | 68k | 37.63 | 50.23 | 46.03 | 6.69 | 8.35 | 5.05 |
| Dynosaur | 66k | **44.35** | 49.34 | 47.08 | 17.26 | 34.59 | 7.11 |
| Muffin (Ours) | 68k | 40.85 | **57.71** | **49.71** | 37.67 | **55.98** | 19.01 |

Table 16: Human evaluation acceptance ratio. We randomly sample 200 instances from each benchmark and let workers evaluate different systems' outputs.

| Models | SuperNI-Test | MMLU | T0-Eval | BBH | Average |
|---|---|---|---|---|---|
| Dolly | 22.5 | 14.0 | 36.5 | 28.0 | 25.3 |
| LongForm | 6.0 | 15.0 | 10.0 | 12.0 | 10.8 |
| Alpaca | 44.5 | 20.0 | 42.0 | 24.0 | 32.6 |
| Alpaca-GPT4 | 45.0 | 11.0 | 38.0 | 24.0 | 29.5 |
| WizardLM | 35.0 | 19.5 | 36.0 | 26.0 | 29.1 |
| Self-Inst. | 39.0 | 23.5 | **45.5** | 29.5 | 34.4 |
| Unnatural Inst. | 50.5 | 24.0 | 34.5 | 23.0 | 33.0 |
| Dynosaur | 43.0 | 28.5 | 30.0 | 22.0 | 30.9 |
| Muffin (Ours) | **56.5** (↑ 6.0) | **34.5** (↑ 6.0) | 45.0 (↓ 0.5) | **31.0** (↑ 1.5) | **41.8** (↑ 7.4) |

Table 17: Pair-wise comparison between MUFFIN (Ours) and three strong baselines, namely SELF-INSTRUCT (Self-Inst.), UNNATURAL INSTRUCT (Unnatural), and SUPERNI, across four benchmarks.

| SuperNI-Test | | | MMLU | | | T0-Eval | | | BBH | | |
|---|---|---|---|---|---|---|---|---|---|---|---|
| Ours | Self-Inst. | Tie | Ours | Self-Inst. | Tie | Ours | Self-Inst. | Tie | Ours | Self-Inst. | Tie |
| **47.0** | 41.5 | 11.5 | **39.5** | 16.5 | 44.0 | **11.0** | 10.0 | 79.0 | **19.5** | 15.5 | 65.0 |
| Ours | Unnatural | Tie | Ours | Unnatural | Tie | Ours | Unnatural | Tie | Ours | Unnatural | Tie |
| **31.5** | 20.0 | 48.0 | **42.5** | 10.0 | 47.5 | **43.5** | 16.5 | 40.0 | **21.5** | 11.5 | 67.0 |
| Ours | SuperNI | Tie | Ours | SuperNI | Tie | Ours | SuperNI | Tie | Ours | SuperNI | Tie |
| **31.0** | 16.0 | 53.0 | **24.0** | 21.0 | 55.0 | 9.0 | **15.0** | 76.0 | **16.5** | 9.5 | 74.0 |

Table 18: Data collection ablations on the validation set of SUPERNI. "CLS Exp" denotes the "classification expansion" (§ 3.3). We fix the data sizes of all the methods to 11.5k.

| Methods | EM (CLS) | ROUGE-L (GEN) | ROUGE-L (overall) |
|---|---|---|---|
| Instruction Rematching | 29.45 | 46.11 | 35.95 |
| Instruction Brainstorm | 28.26 | 47.35 | 36.27 |
| Rematching + Brainstorm | 29.77 | 48.12 | 37.53 |
| Rematching + Brainstorm + CLS Exp | **32.77** | **48.56** | **41.20** |

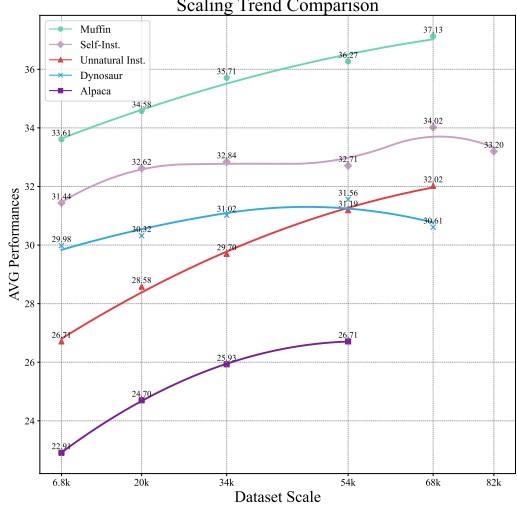

Figure 9: The scaling trends comparison between MUFFIN and the previous baseline datasets (average performances on all four benchmarks).

## K    HUMAN EVALUATION DETAILS

**Acceptance Ratios.**    We employ 5 graduate-level volunteers, who are experienced in NLP and crowd-sourcing jobs, to evaluate different models' outputs. We adopt the blind annotation — each volunteer is randomly responsible for 1 or 2 models' output evaluation without knowing which models they are. When evaluating each instance, we provide the volunteer with only task instruction, input, and model prediction (no ground-truth output is provided to avoid bias in annotation). Then, we ask the volunteer to carefully read the task instructions and decide if the model's output correctly responds to the instruction and input.[25]

**Pair-wise Comparison.**    For each test instance (one task instruction and one input), we provide volunteers with two system outputs, one by our MUFFIN and the other by a baseline dataset. We perform blind annotation on them with random order by asking the volunteers which system's output is preferred in the context of the instruction and the input. The volunteers can also select "tie" if there is no practical quality difference between them.

---

[25]We ask the volunteers to decide the correctness of the models' outputs strictly — considering if the output violates some explicit requirements mentioned in the instruction, e.g., length constraint.

