# OpenReview forum: "MUFFIN: Curating Multi-Faceted Instructions for Improving Instruction Following"
_ICLR.cc/2024/Conference — ICLR 2024 poster_

### Official Review · Reviewer_od6V · 2023-10-30

**Soundness:** 3 good
**Presentation:** 3 good
**Contribution:** 2 fair
**Rating:** 6
**Confidence:** 3

**Summary:**

The paper proposes a new paradigm of "Scaling Tasks per Input" for curating instruction-following datasets. Existing methods for improving language models' instruction-following capabilities have several limitations. Scaling-Inputs require extensive input-output pairs per task, causing oversensitivity to inputs. Scaling Input-Free Tasks expands tasks without inputs but struggles when inputs are needed. To alleviate these issues, this work instead diversifies tasks for each input text using a facet-based approach to enhance model's instruction-following capability.

The authors construct a dataset MUFFIN implementing this paradigm. They generate tasks by identifying textual facets of inputs to stimulate relevant instructions and reusing existing instructions matched to inputs. Careful balancing of classification and generation tasks is also proposed. Comprehensive experiments demonstrate MUFFIN's superiority over strong prior datasets, approaching human-annotated data quality. Ablation studies and human evaluation further verify the effectiveness of the techniques and the overall paradigm.

**Strengths:**

Originality: The proposed `Scaling Tasks per Input` paradigm is novel and unique compared to prior instruction dataset curation schemes. By controlling the input and diversifying tasks, it helps models focus on instructions.

Quality: The paper is technically strong, with a systematic data collection pipeline considering input diversity, instruction generation, output annotation, and task balancing. The analyses thoroughly verify the quality, diversity, and validity.

Significance: Thorough experiments across four benchmarks convincingly demonstrate MUFFIN's superiority, compared to prior synthetic instruction dataset, over strong baselines spanning different paradigms. Both the automatic and human evaluations demonstrated the merit of this dataset.

**Weaknesses:**

While the input-facet-oriented instruction generation approach is interesting, a major concern is the reliability of facets identified by the LLMs. As we know, hallucination remains an issue with LLMs. Though the model may correctly recognize textual attributes in most of time, some "facets" or the model generated *outputs* of these facet-based question could arise from hallucination. I don't see a specific mechanisms to handle this in the paper, except human audit on a small portion of samples. It's unclear whether these noise data, if it indeed exist, would hurt the model's reasoning capabilities. *One potential evidence is the substantially degraded performance on BBH benchmark compared to other baselines in Table 1.*

I see similarities between this work and the self-instruct paper in the overall approach and principles for data generation, experiment design, and analysis approaches. Without addressing the trustworthiness issue I mentioned above, I would be hesitate to advocate the contribution of this work.

I would like to see the authors to share some insights about the limitation of this work.

**Questions:**

1. In the facet-based instruction generation and output annotation, is there a specific strategy or mechanism used to guarantee the trustworthy of the facets or output annotation, except for human inspection?
2. For facet-based generation, how is facet diversity quantified? And is this diversity measured with respect to ground truth facets? Some analysis confirming facet coverage would be reassuring.
3. Could you share some insights about why the MUFFIN model performs much worse than other baselines on BBH benchmark in Table 1?Since BBH focuses on assessing reasoning skills, this seems to indicate there is still room for improvement on those capabilities. To provide more insights into the diversity of the dataset, it would be helpful to breakdown the facet into fine-grained categories, such as elaboration,  numeric calculation, and commonsense reasoning etc. By doing so, it could give a clearer picture of the diversity of this dataset, meanwhile also help to isolate the reason behind the degraded performance.
4. The motivation behind balancing classification and generation tasks is clear, but how exactly is the decision made on which tasks undergo classification expansion? Prompting the model to make the decision?

---

> ### Author Response · Authors · 2023-11-19
> **Author Reponse to Reviewer#4 --- Part 1 (Q1)**
>
> We are grateful for your detailed review and constructive comments! We're pleased you found our work novel, the proposed technique strong, and the experiments solid.
>
> Some of your suggestions are helpful and have been taken by us seriously. In this response, we add more experiments and try to address your concerns one by one:
>
> ---
>
> > Q1: Are there any specific mechanisms to guarantee the trustworthiness of the facets and output generation?
>
> Thanks for pointing this concern out.
>
> 1. First, for the output annotation, we do admit that there is no specific mechanism to guarantee trustworthiness (except human verification), which is a common headache that almost all of the previous works suffered from. However, since the **main objective of this paper is to prove the effectiveness of the proposed paradigm** instead of simply achieving SOTA with a higher-quality dataset, we try to avoid any sophisticated strategies to cherry-pick training instances (e.g., using a small neural model to help filter hallucinated outputs). But of course, your suggestion is highly appreciated; it shall be our next work to release an even higher-quality dataset with the proposed paradigm (e.g., human-annotated Muffin-v2).
>
> 2. As for the facets generation, since it is one of the **unique designs and critical steps** for creating the dataset with the proposed paradigm (i.e., `Scaling Task per Input`), we have some simple yet effective operations in our framework to improve its reliability:
>
>       - **Explicit Prompting**: we added some explicit constraints in the prompt of facets generation, to ensure the quality and diversity of the facets, e.g., “*each facet should clearly describe one specific textual attribute…*” and “*instructions should strictly be based on the facet…*” (pls see Table 4 and Table 5 in Appendix C).
>       - **Joint Filtering**: since the generated facets are eventually used for creating instructions, the proposed “*Instruction Filtering*” can also jointly filter those hallucinated facets that may lead to unanswerable instructions (pls see the second paragraph at page 4).
>
>      We proved the effectiveness of the above methods in Q2.
>
> ---

---

> ### Author Response · Authors · 2023-11-19
> **Author Reponse to Reviewer#4 --- Part 2 (Q2)**
>
> > Q2: This work lacks a clear big picture of the generated facets (the reliability and diversity of facets should be quantified); showing the facets with fine-grained categories may help.
>
> We highly agreed with your concern and realized the importance of showing the quantification of generated facets. We carefully took your comments and tried to address your concern by elaborating on the following three dimensions: **statistics**, **quality**, and **diversity**.
>
> ---
>
> 1. **Facets Statistics**.
>
> We hope the most straightforward way to show a big picture of the generated facets is to illustrate its statistics. As shown below, similar to Table 9 in our paper, we report additional statistics related to facets (the statistics after conducting “*Instruction Filtering*”).
>
> | statistics |  |
> |---|---|
> | # of inputs | 1,463 |
> | # of instructions (from facets-based   brainstorm) | 33,720 |
> | # of facets | 11,382 |
> | avg # of facets per input | 7.78 |
> | max # of facets per input  | 30 |
> | min # of facets per input  | 2 |
> | avg # of instructions per facet | 2.96 |
> | max # of instructions per facet  | 18 |
> | min # of instructions per facet  | 1 |
>
> ---
>
> 2. **Factes Quality**.
>
> To estimate the quality and reliability of the generated facets, and also to demonstrate the effectiveness of our facets-controlling methods (in Q1), we conducted further human verifications on the facets.
>
> Specifically, we randomly collected 100 inputs from Muffin, along with their corresponding facets (889 facets in total) and instructions (2,265 instructions in total). Then, we asked an experienced human annotator to evaluate the following two metrics:
>
>   - **Input-to-facet correctness**: whether each facet correctly described the given input (reflecting the quality and reliability of facts).
>   - **Facet-to-instruction correctness**: whether the subsequent instructions are reasonably related to the given facet (reflecting the utility of facets).
>
> The final results are shown below:
>
> |   input-to-facet correctness   |   90.78%  |
> |---|---|
> |   **facet-to-instruction correctness**  |   **85.22%**  |
>
> These results demonstrate the good quality and utility of the facets generated by our method. It's also worth noting that, after we conducted "*Instruction Filtering*", the input-to-facets correctness increased from 83.28% to 90.78% (the score above), proving the effectiveness of "*Instruction Filtering*" that can jointly filter a considerable amount of hallucinated facets.
>
> ---
>
> 3. **Facets Diversity**.
>
> As for the diversity, we are sincerely grateful for your suggestions. Inspired by your comments, we asked the human annotator to **discover** the categories from the facets (i.e., clustering instead of classification), because it's hard to predefine the categories that may introduce personal bias.
>
> We used the same instances as in the previous analysis (100 inputs and 889 facets). The human annotator was asked to summarize a "keyword" (category) for each LLM-generated facet. If the current category has already appeared before, the annotator also has to "cluster" those facets that belong to the same category together. In doing so, more and more novel categories can be discovered.
>
> We then calculated the following two metrics:
>
> - **Intra-diversity**: how many unique categories there were for each input’s facets, and reported an averaged ratio among all the inputs.
> - **Inter-diversity**: overall categories distribution among all the 889 facets, namely the global frequency of each unique facet.
>
> We found that the generated facets achieve **91.55%** intra-diversity, meaning the facets of the same input are pretty diverse.
>
> Meanwhile, there are a total of **276** unique categories out of the 889 facets, which implies a high inter-diversity of the facets. The distribution plot of inter-diversity with detailed categories can be found at this **[Anonymous Link](https://anonymous.4open.science/api/repo/iclr2024_rebuttal-515F/file/inter_distribution.pdf)**.
>
> Note that the human-annotated facet categories cover a wide range of topics. Some facet categories may describe the structural attributes of the input (e.g., length, conversation format), while others may focus on the specific textual content (e.g., the input contains a location or time entity).
>
> We will update these analyses in our next version and will try to elaborate on more details.

---

> > ### Author Response · Authors · 2023-11-19
> > **Author Reponse to Reviewer#4 --- Part 3 (Q3~Q4)**
> >
> > > Q3: The performance explanation on BBH.
> >
> > We agree that the performances on BBH should be explained more clearly.
> >
> > First, we think it is hard to say “*MUFFIN performs much worse than other baselines on BBH*”, because **only Self-Instruct’s performance is higher than Muffin on BBH**. It is worth noting that the Self-Instruct’s performance is even much better than the human-crafted SuperNI, which is weird since Self-Instruct is generated by GPT-3.
> >
> > 1. One potential perspective to explain Self-Instruct’s high performance is the **updated task leakage analysis** (as suggested by Reviewer#3), where we use ChatGPT to calculate the task category overlap between training datasets and evaluation benchmarks. The **Self-Instruct seems to have a higher task leakage w.r.t. the BBH** (5.29%), though it is just 1%~3% higher than other datasets.
> >
> > 2. Another possible reason is the **different LLMs were adopted** because the main difference between Self-Instruct and Alpaca is the APIs used for creating the datasets (Self-Instruct used GPT-3, while Alpaca utilized InstructGPT_003), while Alpaca didn’t achieve such superior performance on BBH. The instruction and output length distribution of Figure 5 in the [Self-Instruct paper](https://arxiv.org/pdf/2212.10560.pdf) reflects the brief nature of GPT3-generated data [1], while almost all the other datasets have longer and diverse length distribution (especially for those ChatGPT-generated data, as mentioned by Wang et al. [2]; also see the Figure3 in Appendix D of our paper). It may suggest some **superficial overlap** (e.g., similar response length) between Self-Instruct and BBH, because BBH contains substantially short-response tasks (e.g., boolean expression, causal judgment).
> >
> > ---
> >
> > > Q4: How do you decide which tasks undergo classification expansion?
> >
> > Actually, **we didn't use any particular method to decide which task should be expanded**, because any generation task can be intuitively reformulated into a classification structure. The only practical restriction is that some long-text generation tasks may not be suitable for classification expansion (we want the classification options to be short). Therefore, we only exclude the generation tasks with outputs' lengths exceeding a threshold (set at 100 words in this work) while all the other tasks are used for further expansion. We will highlight this point in our next version.
> >
> > ---

---

> ### Author Response · Authors · 2023-11-19
> **Author Reponse to Reviewer#4 --- Part 4 (Q5~Q6)**
>
> > Q5: Comparing this work with the previous works (e.g., self-instruct), it's hard to identify the differences.
>
> We highlight the differences between this work and the previous works below:
>
> - **Motivation**.
>
> The main motivation of almost all previous works is the same — *how to use LLMs to collect a large-scale instruction-following dataset automatically*.
>
> However, the essential motivation of this work is totally different — instead of simply bootstrapping more data, our research question is ***how to reformulate the current learning paradigm for a better instruction-following capacity of LMs***.
>
> We first thought deeply about the scaling-input paradigm; it follows a conventional multi-task learning scheme, where each task instruction has multiple input-output pairs. We assumed that there is a potential harm: models are conditioned on the **same instruction** but **different inputs** to generate **different outputs**, which implicitly teaches models to focus on inputs rather than instruction (because the inputs seem to be the key information to generate correct outputs in this case).
>
> Naturally, we came up with the proposed paradigm, where models are conditioned on **different instructions** but the **same input** to generate totally **different outputs**.
>
> In a word, we respectfully disagree with the opinion that the proposed paradigm is merely a simple incremental production from the previous work. In contrast, *we believe it aligns well with human intuition and can inspire future work in this area*.
>
>
> - **Method**
>
> Following the aforementioned motivation, our data curation method is also quite different from the previous works due to the **unique challenge** in our paradigm.
>
> Most of the previous works tried to use existing instructions (`I`) as demonstrations and asked LLMs to brainstorm new instructions (`I’`), then created the subsequent input (`X`) and output (`Y`). The whole procedure can be succinctly described as `I ⇒ I’ ⇒ (X,Y)`.
>
> Contrastingly, we start our data curation procedure from the input text (`X`) and try to gather diverse instructions (`I’`) for it, and then annotate the outputs (`Y`), namely `X ⇒ I’ ⇒ Y`.
>
> In addition to the different procedures, creating instructions from the input (`X ⇒ I’`) is much more challenging than generating from existing instructions (`I ⇒ I’`), because the former requires more restrictions (instructions should “match” the input, there is less freedom) and is also much harder to be controlled (due to the hallucination of the LLMs). That’s why we proposed two different methods for curating enough high-quality instructions, and used input-oriented facets to guide the instruction brainstorming.
>
> To the best of our knowledge, due to the *unique nature* of the proposed paradigm, *collecting the corresponding dataset requires more special treats and designs in the method*, which any previous work has never covered.
>
> ---
>
> > Q6: Share the limitations of this work.
>
> We are willing to shed light on the following two limitations of this work:
>
> 1. Since our initial motivation is to drive LMs to learn to follow the instructions instead of mapping inputs to outputs (as mentioned before), the contribution of this work mainly focuses on the changes in the dataset paradigms. However, we still follow a traditional supervised fine-tuning scheme to train the LMs. Therefore, **it remains unclear if our paradigm shift can really make a difference during the learning procedure** (such as the difference in gradient). But of course, it's sometimes difficult for humans to interpret the "learning of models". Therefore, **a special learning objective designed for “`Scaling Task per Input`” may help to enhance our conclusion**.
>
> 2. What's more, this work is still following a dataset-scaling scheme in this area; considering the recent findings on "less is more" [3], we think it's also worthwhile to investigate **whether we do need to "`scaling task per input`" or just try to sample "`less task per input`"** and achieves more efficient instruction learning.
>
> Regarding the above limitations, there could be considerable space in this area for the future to dig deep into.
>
> ---
>
> ### References:
> [1]. [Self-Instruct: Aligning Language Models with Self-Generated Instructions.](https://arxiv.org/abs/2212.10560) (*ACL 2023*)
>
> [2]. [How Far Can Camels Go? Exploring the State of Instruction Tuning on Open Resources.](https://arxiv.org/abs/2306.04751) (*NeurIPS 2023 Datasets and Benchmarks*).
>
> [3]. [Lima: Less is more for alignment.](https://arxiv.org/abs/2305.11206) (*ArXiv 2023*)

---

> ### Author Response · Authors · 2023-11-22
> **Kind Reminder from Author**
>
> Dear Reviewer#4 (od6V),
>
> ---
>
> We are writing to inquire if you have had the chance to review our detailed response to your comments. We greatly value your feedback and would appreciate any further questions or comments you might have.
>
> ---
>
> Sincerely,
>
> Authors of Paper#6820

---

> > ### Comment · Reviewer_od6V · 2023-11-23
> >
> > I appreciate the author's detailed clarification and efforts to address my concerns. As all my raised issues have been satisfactorily resolved, I have adjusted my score accordingly. Looking forward to reading the final manuscript.

---

### Official Review · Reviewer_j2qL · 2023-10-31

**Soundness:** 3 good
**Presentation:** 3 good
**Contribution:** 3 good
**Rating:** 6
**Confidence:** 2

**Summary:**

This paper introduced a two-step facet based instruction brainstorming method: in the first step, they use ChatGPT to recognize the textual facets of a given input; in the second step, they instruct ChatGPT to brainstorm task instructions. Besides direct instruction synthesis, the paper also suggests extracting human-written instructions from the training set and employ GPT-4 for binary classification. GPT-4 discriminates if the instruction can align with the input to create a valid task.

**Strengths:**

- The paper contributes to a important problem: dataset and alignment. The paper is well written.

- The proposed method works as a reasonable approach to create a diverse tasks and instructions per task given the same input. GPT-4 works as a reasonable critique model, selecting high quality examples. Overall, the method is very simple but effective.

- The paper reported strong empirical results on SuperNI-test, MMLU, T0-Eval, etc., compared to related work. Human evaluation acceptance ratio is significantly improved with Muffin.

**Weaknesses:**

- The work can have better ablation on whether the diversity in the dataset or the quality of the dataset helps more. For example, without the facet based brainstorming, how would it perform? On the other side, without the filtering (GPT4 critique) how would it perform?

- The paper lacks a scaling analysis on the size of the finetuning dataset. How does the model performance scale with the number of examples in the dataset?

- The discussion on task leaking is too short. I feel it is a very important problem to discuss in greater details. For example, what is the SoTA metric to detect data leaking? Why would semantic similarities be sufficient in detecting? Sentences can mean similarly while being semantically different. I feel the paper can be much stronger if this part is well addressed.

**Questions:**

1. Are there any particular reasons that different GPT models are selected for different purposes? Why would you pick ChatGPT over GPT4 to work on the facet recognition and instruction brainstorm?

2. Any particular reason not to compare with FLAN [1]?

[1]: https://arxiv.org/abs/2301.13688

---

> ### Author Response · Authors · 2023-11-19
> **Author Reponse to Reviewer#3 --- Part 1 (Q1~Q4)**
>
> Thanks for your efforts in reviewing! We are happy you found our paper well-written, the proposed method reasonable, and the empirical results solid.
>
> Some of your suggestions are very important to improve our work. We adopted your comments and tried adding more analyses during this discussion period.
>
> As shown below, we'd like to address your concerns in detail.
>
> ---
>
> > Q1: A better ablation on the trade-off between diversity and quality.
>
> This is a great point for better evaluating the proposed dataset. Your suggestion on the ablation between “facets-based instruction brainstorm” and “GPT-4 based rematching” is a good perspective to investigate this question.
>
> We put the ablation results in Table 13 in Appendix M, where we tested the ablation performances on the development set of SuperNI to avoid cherry-picking (same as the previous work [4]). Here, for your convenience, we also put that table below.
>
> | Methods | EM (CLS) | ROUGE-L (GEN) | ROUGE-L (overall) |
> |---|---|---|---|
> |        Instruction Rematching  | 29.45 | 46.11 | 35.95 |
> |        Instruction Brainstorm  | 28.26 | 47.35 | 36.27 |
> |        Rematching + Brainstorm  | 29.77 | 48.12 | 37.53 |
> |        Rematching + Brainstorm + CLS Expansion  | **32.77** | **48.56** | **41.2** |
>
> We found that **both these methods contribute to the final performance, and seem to show complementary effects on each other** (combining the two methods resulting in better performance).
>
> This conclusion aligns with previous works [1][2], that both factors matter greatly in the final generalization of LMs. However, to our knowledge, it’s still hard to quantify their exact weights because various variables can affect the conclusion (e.g., the base models and the downstream generalization tasks).
>
> We will shed more light on this question in our next version.
>
> ---
>
> > Q2: The paper lacks scaling analysis of the datasets.
>
> Thanks for your reasonable suggestions. It is critical to investigate the scaling trends to prove the robustness of our dataset.
>
> We plotted the performance trends of different datasets (including Muffin and those competitive baselines) with various scales (10%, 30%, 50%, 80% 100%).
>
> For your convenience, **you can find the resulting figure at this** **[Anonymous Link](https://anonymous.4open.science/api/repo/iclr2024_rebuttal-515F/file/scaling_trend_comparison.pdf)**.
>
> The results demonstrate that Muffin consistently achieves superior performances among various scales, and exhibits a stable performance boosting w.r.t. the scaling. However, for some other baseline datasets, the performance even decreased after a certain scale threshold (perhaps due to the effect of the noise).
>
> ---
>
> > Q3: Insufficient discussion on the task leakage.
>
> We do agree with your concern about the task leakage analysis.
>
> To our knowledge, since there is no SOTA leakage detection method specifically designed for instruction tuning, we followed the previous work [3] **using ChatGPT to judge the task leakage of different training datasets**.
>
> Given an instance from a training dataset, we asked ChatGPT whether it belongs to any task categories from the evaluation benchmark. For example, the SuperNI-Test has 12 task categories, and we requested ChatGPT to perform a 13-way classification (one “`None of above`” label) on the training instances. Finally, we report how many instances from this training dataset contain task leakage.
>
> The following table illustrates the results, comparing Muffin with the *most competitive baseline datasets*. Though there are some shifts in the leakage ranking compared with our original analysis (Figure 6 in Appendix N), the overall conclusion is still the same — **Muffin demonstrates relatively low evaluation task leakage across all four benchmarks**.
>
> |  | SuperNI-Test | MMLU | T0-Eval | BBH |
> |---|---|---|---|---|
> | Muffin | 16.01% | 13.05% | 8.63% | 2.53% |
> | Self-Inst. | 11.62% | 10.37% | 11.27% | 5.29% |
> | Unnatural Inst. | 22.27% | 16.91% | 18.39% | 4.76% |
> | Dynosaur | 12.08% | 13.49% | 15.20% | 1.27% |
>
> The main objective of our analyses is to address the impact of other factors that may affect the contribution of our proposed paradigm. Though these leakage analyses are not that perfect, **we believe the task leakage is not a serious issue affecting our conclusion**, according to the above consistent observations.
>
> ---
>
> > Q4: Particular reasons that different GPT models are selected for different purposes.
>
> During our preliminary trials, we observed GPT -4's superiority over ChatGPT in "*Instruction Rematching*", which can produce much more precise pair matching than ChatGPT. While for "*Instruction Brainstorm*", we found that ChatGPT was good enough and much cheaper. That's why we use ChatGPT for "*Instruction Brainstorm*" and GPT-4 for "*Instruction Rematching*". We will highlight this point in our next version.
>
> ---

---

> > ### Author Response · Authors · 2023-11-19
> > **Author Reponse to Reviewer#3 --- Part 2 (Q5)**
> >
> > > Q5: Particular reason not to compare with FLAN.
> >
> > Sorry for letting you get confused.
> >
> > The results of FLAN can be found in the “Existing Systems” of Table 1 of our paper. We didn’t directly compare with FLAN because **it has been trained on some of the evaluation benchmarks** (such as SuperNI-Test), as mentioned by [4], which will result in an unfair comparison.
> >
> > ---
> >
> > ### References:
> > [1]. [How Far Can Camels Go? Exploring the State of Instruction Tuning on Open Resources.](https://arxiv.org/abs/2306.04751) (*NeurIPS 2023 Datasets and Benchmarks*).
> >
> > [2]. [Is prompt all you need? no. A comprehensive and broader view of instruction learning.](https://arxiv.org/abs/2303.10475) (*ArXiv 2023*).
> >
> > [3]. [Dynosaur: A Dynamic Growth Paradigm for Instruction-Tuning Data Curation.](https://arxiv.org/abs/2305.14327) (*EMNLP 2023*).
> >
> > [4]. [Unnatural instructions: Tuning language models with (almost) no human labor.](https://aclanthology.org/2023.acl-long.806.pdf) (*ACL 2023*).

---

> ### Author Response · Authors · 2023-11-22
> **Kind Reminder from Author**
>
> Dear Reviewer#3 (j2qL),
>
> ---
>
> We are writing to inquire if you have had the chance to review our detailed response to your comments. We greatly value your feedback and would appreciate any further questions or comments you might have.
>
> ---
>
> Sincerely,
>
> Authors of Paper#6820

---

> > ### Comment · Reviewer_j2qL · 2023-12-01
> > **Thank you for the response.**
> >
> > Most of my questions are addressed by the rebuttal. Thank you for the timely responses.

---

### Official Review · Reviewer_cEte · 2023-11-01

**Soundness:** 3 good
**Presentation:** 3 good
**Contribution:** 3 good
**Rating:** 5
**Confidence:** 4

**Summary:**

The paper proposes to use multiple facets of an input to construction instruction-input-output pairs.

**Strengths:**

1. Curating instructions from different facets is an interesting idea. It can be combined with other data augmentation techniques.
2. The presentation is straightforward and easy to follow.

**Weaknesses:**

1. This method heavily relies on the human-curated SuperNI dataset. The Instruction Brainstorming requires extracting Instruction-input pairs from SuperNI while instruction reconstruction simply use data from SuperNI but do compositional changes. However, after those changes, the results of finetuning an LLM become worse than originally finetuning with SuperNI, as in Table 1. Does this mean we are introducing noise?
2. Comparisons to some baselines might not be fair, although being claimed to directly compared results in the paper. I tihink the comparison with Dynasour and Unnatural instruction is unfair as again, the curation of the new dataset relies on some higher quality human instruction in SuperNI while the other two work doesn't rely on modifying human instructions. I think to really demonstrate the effectiveness of this input-based instruction construction method, the authors should try to start from Dynasour or Unnatural instruction and create new instructions on inputs from those datasets instead of from SuperNI.
3. The creation of the facets lack human control. This is a limitation in the methodology that the framework cannot control which facets will be generated from the first stage.
4. This might be personal, but I have an overall negative attitude towards work like this one. This work basically falls into the category of exploiting the ability of LLMs to augment the instruction tuning data. The changes in this paradigm compared to previous ones look incremental, simply trying to change a bit on how we prompt LLMs to generate instruction-input-output pairs. Notice that this idea is also not new as in Dynasour, they also generate multiple instructions by asking LLMs to focus on different parts of the metadata. Most importantly, I don't know how this paradigm would really impact how people create instruction data. Does it worth trying this paradigm or simply using the queries to chatgpt to collect more data?

**Questions:**

1. See the weakness section. Maybe elaborate more on your opinions about point 1 and 2.

---

> ### Author Response · Authors · 2023-11-19
> **Author Reponse to Reviewer#2 --- Part 1 (Q1~Q4)**
>
> Your comments are very much appreciated! We are glad you found our idea interesting and the presentation clear.
> We took your comments carefully and have added some missing experiments you mentioned.
> We address your concerns one by one as follows:
>
> ---
>
> > Q1: Muffin highly relies on the human-crafted SuperNI dataset.
>
> Our data curation method does utilize the resources from SuperNI, but it’s hard to say our method “*highly*” relies on SuperNI, because:
> - For “*Instruction Brainstorm*”, we only used 3 instructions from SuperNI as demonstrations (which is similar to all the previous works); For “*Instruction Rematching*”, we extracted instructions and inputs from SuperNI **separately while NOT keeping the original correspondence** between them (the instruction-input correspondence should be the most critical part of human annotation).
> - Additionally, our method can be practically applied to **any** textual resources and still results in comparable instruction-following performances (please refer to the *experimental results in Q3*).
>
> ---
>
> >Q2: Muffin is sourced from SuperNI, but introduces more noise.
>
> We agree with your concern and address it with the following points:
> - To our knowledge, all the LLM-generated datasets (e.g., alpaca, self-inst) introduced more noise compared with the source datasets. However, one main contribution of this area is to use LLMs to automatically curate instruction-following instances instead of relying on expensive human labor.
> - According to our human analysis, Muffin suffers less noise than previous works (pls see Figure 4 in Appendix E).
>
> ---
>
> > Q3: The authors should create new instructions on the inputs sourced from Dynosaur or Unnatural Inst. to ensure a fair comparison.
>
> This is a very good suggestion!
>
> During this discussion period, we gathered the inputs from other LLM-generated datasets (including Dynosaur and Unnatural-Inst, as suggested by you), and applied our “*Instruction Brainstorm*” methods to them. Finally, we created two small-scale datasets (about 16k), namely *Muffin-Dynosaur* and *Muffin-Unnatural*, and compared their performances with Dynosaur and Unnatural-Inst (in the same size).
>
> The results (with T5-3B models) can be found in the following table:
>
> | **Models** | **SuperNI (CLS)** | **SuperNI (GEN)** | **SupnerNI (overall)** | **MMLU (ACC)** | **MMLU (EM)** | **T0-Eval (ACC)** | **T0-Eval (EM)** | **BBH (EM)** | **AVG** |
> |---|:---:|:---:|:---:|:---:|:---:|:---:|:---:|:---:|:---:|
> | Unnatural Inst. | 23.94 | 44.36 | 37.12 | 24.35 | 22.33 | 45.93 | 35.88 | 8.69 | 30.33 |
> | Dynosaur | 25.03 | 42.08 | 32.55 | 26.88 | 25.26 | 38.56 | 39.13 | 12.09 | 30.2 |
> | Muffin-Unnatural | 27.53 | 47.7 | 38.86 | 33.12 | 24.73 | **46.08** | 43.63 | 12.91 | 34.32 |
> | Muffin-Dynosaur | 27.72 | 46.7 | 38.23 | **33.34** | 22.59 | 45.3 | **44.01** | **13.53** | 33.93 |
> | Muffin | **30.77** | **48.82** | **40.45** | 33.24 | **25.39** | 42.68 | 40.99 | 13.44 | **34.47** |
>
> ---
>
>  We observed that Muffin consistently demonstrates superior generalization performance, regardless of the text sources employed. It further suggests that Muffin's robust ability to follow instructions is not reliant on the input resources but rather **stems more from our crucial contribution—the diversified instructions per input**. Therefore, we anticipate that our method can be ideally applied to any resources while still crafting diverse task instructions.
>
> ---
> > Q4: Is it worth trying this paradigm or simply using the queries to chatgpt to collect more data?
>
> We hope we can answer your question by plotting the scaling trends of different datasets. Specifically, we randomly sampled subsets from the original datasets and trained T5-3B on them to show the trends (10%, 30%, 50%, 80%, 100%).
>
> The resulting figure can be found in this **[Anonymous Link](https://anonymous.4open.science/api/repo/iclr2024_rebuttal-515F/file/scaling_trend_comparison.pdf)**.
>
> In the range of 68K, MUFFIN exceeds the baselines by a noteworthy margin (average scores on 4 evaluation benchmarks). Other baselines may only **be comparable to our data results when they continue to be scaled to several times the size of our data**.
>
> More importantly, the performances of some datasets even decrease after scaling to a larger size (perhaps due to the noise in these LLM-generated datasets), such as Self-Inst. and Dynosaur. Therefore, we conjecture that our paradigm is more efficient than simply collecting more data from the other paradigms.
>
> ---

---

> > ### Author Response · Authors · 2023-11-19
> > **Author Reponse to Reviewer#2 --- Part 2 (Q5)**
> >
> > > Q5: The creation of facets lacks human control.
> >
> > This is a good point. It is true that we do not have a perfect solution to solve this problem, but this work nonetheless employed the following simple and effective methods:
> > - In our data generation prompt, we added some **explicit constraints** when asking LLMs to generate facets, to ensure the quality and diversity of the facets (pls see Table 4 and Table 5 in Appendix C).
> > - The generated facets are eventually used for creating instructions. Therefore, the proposed **“*Instruction Filtering*”** can considerably filter those hallucinated facets that may lead to unanswerable instructions (pls see the second paragraph on page 4).
> >
> >
> > To quantify the facets and demonstrate the effectiveness of the above methods, as suggested by *Reviewer#4*, we conducted two further small-scale human verifications on the generated facets.
> >
> > 1. **Quality Verification**.
> >
> > We randomly collected 100 inputs with their LLM-generated facets (889 facets in total) and instructions (2,265 instructions in total), and then asked a human annotator to decide whether each facet correctly describes the given input (**input-to-facet correctness**). In addition, we also investigated whether the subsequent instructions are reasonably related to the given facet (**facet-to-instruction correctness**).
> >
> > We found that **90.78%** of input-facet pairs are correct, and **85.22%** of facet-instruction pairs are valid, demonstrating a good quality and utility of the facets generated by our method.
> >
> > It's also worth noting that the **input-to-facets correctness increased from 83.28% to 90.78% after we conducted "*Instruction Filtering*"**, implying our framework can filter a considerable amount of hallucinated facets (which *proves the effectiveness of the aforementioned second method*).
> >
> > 2. **Diversity Verification**.
> >
> > We also asked the human annotator to tag a fine-grained category for each facet to show the diversity. Specifically, the annotator was asked to summarize a “keyword” for each facet as the category and figure out which facets belong to the same category (similar to “clustering”).
> >
> > We calculated how many unique categories there were for each input’s facets, and reported an averaged ratio among all the inputs (**intra-diversity**). Meanwhile, we also showed the overall categories distribution, namely the diversity of unique categories (**inter-diversity**).
> >
> > The generated facets achieve **91.55%** intra-diversity, meaning the facets of the same input are pretty diverse. Meanwhile, there are a total of **276** unique categories out of the 889 facets (100 inputs), which implies a high inter-diversity of the facets.
> >
> > For your convenience, we provide the plot of inter-diversity at this **[Anonymous Link](https://anonymous.4open.science/api/repo/iclr2024_rebuttal-515F/file/inter_distribution.pdf)**. Note that the human-annotated facet categories cover a wide range of topics. Some facet categories describe the structural attributes of the input (e.g., length, conversation format), while others focus on the textual content (e.g., the input contains a specific location or time).

---

> > > ### Author Response · Authors · 2023-11-19
> > > **Author Reponse to Reviewer#2 --- Part 3 (Q6)**
> > >
> > > > Q6: The changes in this paradigm compared to previous ones look incremental, simply trying to change a bit on how we prompt LLMs to generate instruction-input-output pairs.
> > >
> > > Thank you for pointing this concern out!
> > >
> > > To our knowledge, this work is essentially different from the previous works, including the motivation of the proposed paradigm and the data curation method.
> > >
> > > ---
> > >
> > > - **Motivation**.
> > >
> > > The main motivation of almost all previous works is the same — *how to use LLMs to collect a large-scale instruction-following dataset automatically*.
> > >
> > > However, the essential motivation of this work is totally different — instead of simply bootstrapping more data, our research question is ***how to reformulate the current learning paradigm for a better instruction-following capacity of LMs***.
> > >
> > > We first thought deeply about the scaling-input paradigm; it follows a conventional multi-task learning scheme, where each task instruction has multiple input-output pairs. We assumed that there is a potential harm: models are conditioned on the **same instruction** but **different inputs** to generate **different outputs**, which implicitly teaches models to focus on inputs rather than instruction (because the inputs seem to be the key information to generate correct outputs in this case).
> > >
> > > Naturally, we came up with the proposed paradigm, where models are conditioned on **different instructions** but the **same input** to generate totally **different outputs**.
> > >
> > > In a word, we respectfully disagree with the opinion that the proposed paradigm is merely a simple incremental production from the previous work. In contrast, *we believe it aligns well with human intuition and can inspire future work in this area*.
> > >
> > > ---
> > >
> > > - **Method**
> > >
> > > Following the aforementioned motivation, our data curation method is also quite different from the previous works due to the **unique challenge** in our paradigm.
> > >
> > > Most of the previous works tried to use existing instructions (`I`) as demonstrations and asked LLMs to brainstorm new instructions (`I’`), then created the subsequent input (`X`) and output (`Y`). The whole procedure can be succinctly described as `I ⇒ I’ ⇒ (X,Y)`.
> > >
> > > Contrastingly, we start our data curation procedure from the input text (`X`) and try to gather diverse instructions (`I’`) for it, and then annotate the outputs (`Y`), namely `X ⇒ I’ ⇒ Y`.
> > >
> > > In addition to the different procedures, creating instructions from the input (`X ⇒ I’`) is much more challenging than generating from existing instructions (`I ⇒ I’`), because the former requires more restrictions (instructions should “match” the input, there is less freedom) and is also much harder to be controlled (due to the hallucination of the LLMs). That’s why we proposed two different methods for curating enough high-quality instructions, and used input-oriented facets to guide the instruction brainstorming.
> > >
> > > To the best of our knowledge, due to the *unique nature* of the proposed paradigm, *collecting the corresponding dataset requires more special treats and designs in the method*, which any previous work has never covered.

---

> > ### Comment · Reviewer_cEte · 2023-11-22
> >
> > I appreciate the authors' efforts in conducting the strategies in this paper on Dynasour and Unnatural-Instruct. The results look good.
> >
> > However, I'm still concerned about how much this method relies on the existing human-annotated datasets. It looks like from Table 13 that the classification expansion is the most effective strategy among the three, which is the reason that differs MUFFIN from other baselines (without it the performance is ~37 which is incrementally better than baselines and much worse than SuperNLI). It has been known that the performance on SuperNLI-test set and other benchmarks like MMLU will be influenced a lot by the number of classification tasks, Adding more classification tasks, in my opinion, is a shortcut to improve the performance. Thus, I'm a bit concerned this method is only designed to improve the performance as it doesn't totally fit into the story. The story is to augment instructions, but classification expansion simply converts some human-annotated instruction-input-output triplets to another format.
> >
> > Besides, I'm still not sure whether this paradigm can be a widely adopted one because of its being hard to control. We will not be able to control which facets to use and how many generation tasks to convert to classification tasks to optimize model performance.
> >
> > So, I prefer not to change my original scores. I'm pretty confidence about the current score.

---

> > > ### Author Response · Authors · 2023-11-22
> > > **Author's Follow-up Response to Reviewer#2**
> > >
> > > Many thanks for sharing your further comments. Regarding your comments respectfully, we want to make the following clarifications.
> > >
> > > ---
> > >
> > > > “I'm still concerned about how much this method relies on the existing human-annotated datasets ... classification expansion simply converts some human-annotated instruction-input-output triplets to another format.”
> > >
> > > First, in this work, **we didn’t use any complete human-annotated instruction-input-output triplets** from SuperNI. In contrast, we only:
> > > 1. Use existing 3-shot instructions as in-context demonstrations. Similar to many of the previous works [1][2][3].
> > > 2. Use existing input texts. As proved by the experiments in A3, our method can utilize any input texts instead of only relying on SuperNI.
> > > 3. Use **shuffled** input-instruction pairs **without outputs**. We broke the correspondence between the origin instruction-input pairs. And, no human-annotated outputs were used here (all the outputs were annotated by ChatGPT).
> > >
> > > Therefore, the extent to which we use SuperNI is similar to previous works. Because input-instruction-output correspondence should be the most critical part of a human-annotated instruction dataset; while our datasets have never taken this kind of information, as we broke the relation of input-instruction and adopted LLM-generated outputs.
> > >
> > > Second, we only used human-written instructions in “*Instruction Rematching*”. For “*Instruction Brainstorm*”, all the instructions were generated by ChatGPT, which is exactly the same as the previous works. While in A3, we have proved that our dataset can still result in superior performances with only LLM-brainstormed instructions.
> > >
> > > ---
> > >
> > > > “Adding more classification tasks, in my opinion, is a shortcut to improve the performance … only designed to improve the performance as it doesn't totally fit into the story.”
> > >
> > > We think it’s hard to say that “adding more classification tasks is a *shortcut* ”:
> > > 1. Almost all the real-world NLP tasks can be roughly divided into generation and classification tasks, where the classification tasks are found to occupy a considerable percentage in real-world applications [4]. Therefore, for creating a multi-task instruction dataset, **it’s the duty to add more classification tasks** to improve the task diversity, which also works for those human-annotated datasets [6][7].
> > > 2. **Adding more classification tasks is also a convention** that has been widely adopted by previous works. For example, Dynosaur [5] sampled classification tasks with a higher probability to enforce models to learn more classification tasks; Self-Instruct [1] and Unnatural Instruct [2] all designed the specific procedures for boosting classification tasks (i.e., classification-label-first generation, classification constraints generation). Therefore, we believe the comparison with baselines is fair to support our paper’s conclusion.
> > >
> > > What’s more, since the main target of this work is to gather different task instructions for each input, collecting completely distinct task instructions is a crucial step for our paradigm (i.e., enforcing models to follow various instructions with the same input). Thus, adding more classification tasks can also **diversify the task objectives** of instructions, aligning with our story.
> > >
> > > ---
> > >
> > > References:
> > >
> > > [1]. [Self-Instruct: Aligning Language Models with Self-Generated Instructions.](https://arxiv.org/abs/2212.10560) (*ACL 2023*).
> > >
> > > [2]. [Unnatural instructions: Tuning language models with (almost) no human labor.](https://aclanthology.org/2023.acl-long.806.pdf) (*ACL 2023*).
> > >
> > > [3]. [Stanford Alpaca: An Instruction-following LLaMA Model.](https://crfm.stanford.edu/2023/03/13/alpaca.html) (*Blog 2023*)
> > >
> > > [4]. [A Universal Discriminator for Zero-Shot Generalization.](https://arxiv.org/abs/2211.08099) (*ACL 2023*)
> > >
> > > [5]. [Dynosaur: A Dynamic Growth Paradigm for Instruction-Tuning Data Curation.](https://arxiv.org/abs/2305.14327) (*EMNLP 2023*).
> > >
> > > [6]. [Super-NaturalInstructions: Generalization via Declarative Instructions on 1600+ NLP Tasks.](https://arxiv.org/abs/2204.07705) (*EMNLP 2022*)
> > >
> > > [7]. [Finetuned Language Models Are Zero-Shot Learners.](https://arxiv.org/pdf/2109.01652.pdf) (*ICLR 2022*)

---

> ### Author Response · Authors · 2023-11-22
> **Kind Reminder from Author**
>
> Dear Reviewer#2 (cEte),
>
> ---
>
> We are writing to inquire if you have had the chance to review our detailed response to your comments. We greatly value your feedback and would appreciate any further questions or comments you might have.
>
> ---
>
> Sincerely,
>
> Authors of Paper#6820

---

### Official Review · Reviewer_BsoU · 2023-11-01

**Soundness:** 3 good
**Presentation:** 2 fair
**Contribution:** 3 good
**Rating:** 8
**Confidence:** 4

**Summary:**

This paper describes a technique for synthesizing instruction fine-tuning data using LLMs (ChatGPT and GPT-4). In particular, the paper draws a distinction between past works in this area which have focused on either adopting an instruction+input format and scaling the number of inputs per instruction (Scaling-Inputs), or adopting an instruction-only format and scaling the number of total instructions (Scaling Input-Free Tasks). As an alternative, the technique and resulting dataset, MUFFIN, presented in this work adopts the instruction+input format but uses automated techniques to scale the number of instructions per input (Scaling Tasks per Input).

Experimental comparisons to extensive baselines are presented on SuperNI, MMLU, T0, and BBH (each classified as either Scaling-Inputs, Scaling Input-Free Tasks, or Hybrid) and demonstrate the effectiveness of the proposed approach in all 3 settings. Additional experiments and results from human evaluation are also presented.

**Strengths:**

1. The topic of how to effectively scale synthetic instruction datasets is relevant and timely, and the dichotomy of scaling inputs vs instructions is interesting.
2. The experiments are extensive and the inclusion of evaluation of datasets from multiple settings (scaling-inputs, input-free, etc) is appreciated.
3. The methodology and techniques are mostly well-described and presented clearly.

**Weaknesses:**

1. The muffin looks more like a cupcake to me.
2. As noted previously, the dichotomy of scaling inputs vs instructions is interesting. However, the presentation makes it seem like one must choose between the two when really they are more like two extremes of a spectrum. This very naturally leads me to wonder whether a hybrid that scales both according to some mixture proportion would be most effective. This seems like a somewhat large omission given it would be very testable with the synthetic data here.
3. All experiments use T5-3B or T5-11B. As many of the other datasets (Self-Instruct, Dolly, Alpaca) were developed and tested using more recent LMs (Llama, etc), it would be useful to see if/how the results change when fine-tuning such models. At minimum, it would be useful to see performance numbers for the original models developed with these datasets (if they are available) in the "Existing Systems" section of Table 1 to understand the performance drop due to changes in setup (model, hyperparams, etc).
4. I think there are a few errors in the baseline discussion in section 5. In particular, this section says Dolly and Self-Instruct were collected with ChatGPT or GPT-4. As far as I know, Dolly is human-created (this is stated correctly in the Appendix) and Self-Instruct was produced using GPT-3.
5. The presentation of human evaluation was somewhat confusing and left out key details. What model was used, 3B or 11B? Additionally, the evaluation is described as follows, "we provide the volunteer with only task instruction, input, and model prediction." However, MMLU is classified as "Input-Free". What do annotators see as the input in this setting?

**Minor Issues**

1. The Scaling-Inputs, Scaling Input-Free Tasks, and Scaling Tasks per Input terminology is somewhat awkward. Have the authors considered something slightly simpler like Scaling-Inputs, Scaling-Input-Free, and Scaling-Instructions?
2. I found the "our MUFFIN" terminology used throughout the paper somewhat annoying and distracting.

------

I have reviewed the authors' responses and raised my score accordingly.

**Questions:**

See weaknesses

---

> ### Author Response · Authors · 2023-11-21
> **Author Reponse to Reviewer#1 --- Part 1 (Q1~Q3)**
>
> Many thanks for your detailed review! We are glad you found this work interesting, the experiments solid, and the presentation clear. During this discussion period, we conducted more experiments per your suggestions to improve this work. As shown below, we summarize your main concerns and address them one by one:
>
> ---
>
> > Q1: Whether a hybrid paradigm that scales both (`Scaling Inputs` & `Scaling Tasks per Input`) according to some mixture proportion would be most effective.
>
> This is a great point to investigate!
>
> To figure out the answer to this question, we mixed Muffin (`Scaling Tasks per Input`) with SuperNI-Train (`Scaling Inputs`) with various proportions (0.0, 0.1, 0.2, …, 0.9, 1.0), where the proportion means how many instances are from Muffin. Then, we trained T5-3B on these mixtures and reported the average performances on the four benchmarks.
>
> For your convenience, we plotted the figure at this **[Anonymous Link](https://anonymous.4open.science/api/repo/iclr2024_rebuttal-515F/file/mixture_trends.pdf)**.
>
> Note that, SuperNI-Train is created by humans and has much higher quality. That’s why using only SuperNI-Train resulted in the best performance in this figure (i.e., `proportion == 0`).
>
> However, the most interesting finding is that, after mixing Muffin with more high-quality data from SuperNI-Train, the **model’s performance first dropped and then increased**.
>
> Thus, we draw mainly two conclusions here:
> 1. **The dataset paradigm does affect the learning efficiency**. Different dataset paradigms do have obviously various impacts on the model’s performance. Therefore, simply combining two different paradigms can even hurt the model’s instruction-tuning efficiency, meaning the paradigm is a critical factor.
> 2. **The superiority of the proposed paradigm**. Meanwhile, the effect of dataset paradigms is even greater than that of data quality (adding a small proportion of high-quality data from other paradigms even harms Muffin’s performance), further implying the proposed paradigm's effectiveness.
>
> This observation also inspires us to release an even higher-quality dataset in the future, that has both quality and paradigm superiority (as also suggested by reviewer#4).
>
> ---
>
> > Q2: The performances with the most recent LLMs (e.g., LLaMA).
>
> We agree that we have to adopt some other models besides T5.
>
> During this discussion period, we conducted experiments with the most advanced decoder-only LMs, namely **LLaMA-2 (13B)**, to further enhance the reliability of our conclusion. Since the training and inference of LLaMA-2-13B were time-consuming, we reported only the performances of Muffin and those most competitive baselines; meanwhile, we randomly selected 2k testing instances from each evaluation benchmark (every evaluation task was covered) to show the comparison, instead of the whole testing benchmarks (which will take more than two weeks).
>
> The results are shown in the following table. The main conclusion is similar to T5’s results in our paper, where Muffin consistently gains better overall performances compared with the baselines.
>
> | Models | SuperNI (CLS) | SuperNI (GEN) | SupnerNI (overall) | MMLU (ACC) | MMLU (EM) | T0-Eval (ACC) | T0-Eval (EM) | BBH (EM) | AVG |
> |---|:---:|:---:|:---:|:---:|:---:|:---:|:---:|:---:|:---:|
> | Self-Inst. | 27.34 | 40.9 | 33.78 | 33.56 | 34.28 | 56.78 | 51.55 | **33.77** | 38.99 |
> | Unnatural Inst. | 38.12 | 46.98 | 45.02 | 38.75 | 27.98 | 50.13 | 49.88 | 13.56 | 38.8 |
> | Dynosaur | **39.13** | 50.09 | 45.84 | 34.37 | 35.82 | 51.28 | 47.2 | 12.33 | 39.51 |
> | Muffin | 38.3 | **53.41** | **46.62** | **46.46** | **43.3** | **60.07** | **52.37** | 27.08 | **45.95** |
>
> We will add the completed inference results of LLaMA-2-13B in our next version, and fill in the performances of the "original models" trained on those baseline datasets to the "Existing Systems" in Table 1.
>
> ---
>
> > Q3: The presentation of human evaluation omits key details.
>
> We are sorry for letting you get confused.
>
> - *For the models used in human evaluation*: We use **T5-3B** through all the analysis sections. We will highlight it by mentioning it in the table captions.
> - *What “input” was provided to the human when facing "input-free" tasks*: For those "input-free" tasks, we just **left the "input" field empty** (pls refer to Table 12 in Appendix L). For a better illustration, we provide the figure of the human evaluation interface at this **[Anonymous Link](https://anonymous.4open.science/api/repo/iclr2024_rebuttal-515F/file/human_eval_interface.jpg)**.
>
> ---

---

> > ### Author Response · Authors · 2023-11-21
> > **Author Reponse to Reviewer#1 --- Part 2 (Q4~Q5)**
> >
> > > Q4: Presentation errors, e.g., Dolly is a human-crafted dataset.
> >
> > You are correct! We will fix this presentation error in our next version.
> >
> > ---
> >
> > > Q5: Awkward terminologies (e.g., `Scaling Tasks per Input`).
> >
> > Thanks for your suggestions!
> >
> > - *For the terminologies of dataset paradigms*: We tried different terms in our earlier draft, such as `Instruction-driven Scaling` or `Input-driven Scaling`, and found those terms were not clear enough to present. Personally, we thought **good terms should help distinguish the key difference among these three paradigms**. Therefore, after careful consideration, we finally decided to use these “straightforward” but clear terms, such as `Scaling Inputs` and `Scaling Tasks per Input`. We will try to figure out better terminologies if possible.
> > - *For the terminology of “our MUFFIN”*: Thanks for sharing your opinion. We will use “MUFFIN” or “our paradigm” instead in our next version.

---

> ### Author Response · Authors · 2023-11-22
> **Kind Reminder from Author**
>
> Dear Reviewer#1 (BsoU),
>
> ---
>
> We are writing to inquire if you have had the chance to review our detailed response to your comments. We greatly value your feedback and would appreciate any further questions or comments you might have.
>
> ---
>
> Sincerely,
>
> Authors of Paper#6820

---

> > ### Comment · Reviewer_BsoU · 2023-11-22
> > **Reply to Authors**
> >
> > Thanks for the thorough reply and addressing my questions. I will focus on Q2 as I feel my other questions have been addressed sufficiently.
> >
> > The findings from the mixture experiment are quite interesting. I appreciate the authors' putting forward some hypotheses, but I'm not sure I can confidently draw the same conclusions (at least not with any confidence). Please ensure any such hypotheses put forward in the paper are clearly noted as such. I am particularly unsure how to interpret that "adding a small proportion of high-quality data from other paradigms even harms Muffin’s performance" as this seems counter to my general intuition.
> >
> > I believe the paper has improved since my initial review and represents a valuable contribution; at minimum bringing attention to a previously unexplored dichotomy in the way instruction examples are formatted, providing a useful instruction tuning data set, and putting forward some interesting results related to the way mixing datasets harms performance. I will raise my score accordingly.

---

> ### Author Response · Authors · 2023-11-23
> **Author's Follow-up Response to Reviewer#1**
>
> Dear Reviewer#1 (BsoU),
>
> ---
>
> Thanks so much for reading our response and raising the score. It's our pleasure that our additional experiments and efforts can earn your recognition.
>
> Regarding the interesting results of mixing SuperNI, we will definitely attach it with more details and try to figure out more reasonable explanations in our next version.
>
> Thanks again for your recognition of our paper's contributions.
>
> ---
>
> Best regards,
>
> Authors of Paper#6820

---

### Meta-Review · Area_Chair_9twA · 2023-12-07

**Metareview:**

This paper presents MUFFIN, a novel scheme for instruction-following dataset curation that addresses the limitations of existing scaling approaches in large language models. The paper is timely and relevant, with an interesting exploration of the dichotomy between scaling inputs and instructions. The experiments are extensive, evaluating multiple dataset settings, and the methodology is well-described and clearly presented. Given the significance of the topic and the comprehensive experimental evaluation, I recommend acceptance of this paper.

**Justification For Why Not Higher Score:**

n/a

**Justification For Why Not Lower Score:**

n/a

---

### Decision · Program_Chairs · 2024-01-16

Accept (poster)